# Long-term changes in the thermodynamic structure of the lowermost stratosphere inferred from reanalysis data

Franziska Weyland[1], Peter Hoor[1], Daniel Kunkel[1], Thomas Birner[2,3], Felix Plöger[4,5], and Katharina Turhal[4,5]

[1]Institute for Atmospheric Physics, Johannes Gutenberg University Mainz, Germany
[2]Meteorological Institute, Ludwig-Maximilians-University Munich, Germany
[3]Institute for Physics of the Atmosphere, Deutsches Zentrum für Luft- und Raumfahrt, Oberpfaffenhofen, Germany
[4]Institute of Climate and Energy Systems, Stratosphere (ICE-4), Forschungszentrum Jülich, Jülich, Germany
[5]Institute for Atmospheric and Environmental Research, University of Wuppertal, Wuppertal, Germany

**Correspondence:** Franziska Weyland (franziska.weyland@uni-mainz.de)

**Abstract.** The lowermost stratosphere (LMS) plays an important role for stratosphere-troposphere coupling and the Earth's radiation balance. This study investigates the effects of long-term changes of the tropopause and the lower stratospheric isentropic structure on the mass of the LMS. We compare five modern reanalyses: ERA5, ERA-Interim, MERRA-2, JRA-55 and JRA3Q. The focus is on changes after 1998, which marks the anticipated beginning of stratospheric ozone recovery. The trend

analysis is performed with a dynamic linear regression model (DLM), capable of modeling non-linear trends.

According to our study, isentropic pressure in the lower stratosphere (here 380–430 K) shows negative trends in the tropics and positive trends in the extratropics. In the Northern hemisphere (NH), we find that the extratropical tropopause is rising, accompanied by decreasing pressure at an average rate of $-1\,\mathrm{hPa/decade}$. Additionally, our results indicate that the tropical tropopause in the NH has expanded polewards by $0.5°$ latitude between 1998–2019. In the Southern hemispheric (SH)

extratropics, the lapse rate tropopause shows a downward tendency of up to $+2\,\mathrm{hPa/decade}$ after 1998, consistent across all reanalyses except JRA3Q. The tropical tropopause and the cold point is rising, accompanied by decreasing pressure at a rate of ca. $-0.5\,\mathrm{hPa/decade}$ in all reanalyses. The sign of the tropical tropopause potential temperature trends, however, differs across the reanalyses. This can be attributed to contrasting (absolute) temperature trends in the tropical tropopause region, such as at the $100\,\mathrm{hPa}$ pressure level.

Consistent with the upward and poleward trend of the NH tropopause, the mass of the LMS is decreasing by $2$–$3\,\%$ between 1998–2019 if a fixed isentrope ($380\,\mathrm{K}$) is chosen as the upper LMS boundary. In ERA5, as well as MERRA-2 and ERA-Interim, this mass decline disappears if dynamical upper LMS boundaries are used, that take the upward trends of the tropical tropopause into account.

## 1 Introduction

The lowermost stratosphere (LMS), also referred to as stratospheric part of the "middle world", is defined as the region of the stratosphere where isentropic surfaces intersect with the tropopause (Holton et al., 1995). The "middle world" can be distin-

guished from the "overworld", where isentropes are situated entirely in the stratosphere, and the "underworld" with isentropes exclusively located in the troposphere (Hoskins, 1991). Along these intersecting isentropes, quasi-isentropic exchange between the tropical troposphere and the extratropical stratosphere is possible. The upper boundary of the LMS has often been approxi-
mated by the 380 K isentrope which, in turn, can be regarded as an approximation of the tropical tropopause (e.g., Appenzeller et al., 1996; Schoeberl, 2004; Olsen et al., 2013; Wang and Fu, 2021).

The LMS includes the extratropical transition layer (ExTL) (WMO, 2003), which has been defined on the basis of trace gas gradient changes in the LMS (e.g. Hoor et al., 2004), consistent with trajectory studies (Berthet et al., 2007). The ExTL is characterized by irreversible mixing across the tropopause and high variability of radiatively relevant trace gases like water
vapor and ozone (Pan et al., 2004; Hoor et al., 2002; Fischer et al., 2000). The ExTL partly coincides with the tropopause inversion layer (TIL) as marked by a maximum of static stability (Birner et al., 2002; Birner, 2006). Further, the ExTL region is characterized by strong shear occurrence just around the extratropical tropopause (Kunkel et al., 2019; Kaluza et al., 2021). The LMS is essential for the Earth's radiation budget (e.g., Forster and Shine, 1997, 2002; Zhang et al., 2004; Riese et al., 2012) since it constitutes a region of strong composition gradients of, e.g., ozone and water vapor. Changes of the LMS com-
position affect local temperature gradients (e.g., Randel et al., 2007) with impacts on climate and circulation (e.g., Hegglin and Shepherd, 2009; Charlesworth et al., 2023). Furthermore, the LMS contains a significant proportion of stratospheric mass, and thereby a considerable part of total column ozone, due to the exponential density decrease with altitude. From Fig. 5 in Appenzeller et al. (1996), the LMS mass can be estimated to about 30–50 % of the total stratospheric mass. LMS and strato-spheric mass show a seasonal cycle, closely following the seasonal cycle of tropopause height (Appenzeller et al., 1996). On
short time scales, LMS mass variations are linked to stratospheric-tropospheric mass exchange (e.g., Stohl, 2003; Schoeberl, 2004; Hegglin et al., 2010; Škerlak et al., 2014). This affects the abundance of trace gases such as ozone, which show sharp gradients in the UTLS (Upper Troposphere - Lower Stratosphere). Due to this transport barrier property, which gives rise to strong trace gas gradients, the tropopause location is of special importance.

The LMS is bounded below by the extratropical tropopause. The location of the (extratropical) tropopause can be defined thermally based on the temperature lapse rate (WMO, 1957) or the potential temperature gradient (Tinney et al., 2022), dy-namically by choosing a characteristic value of potential vorticity (PV) (e.g., Hoerling et al., 1991) or by PV gradients (Kunz et al., 2011), and chemically via trace gas gradients (e.g., ozone) or trace gas correlations (e.g., Bethan et al., 1996; Fischer et al., 2000). The tropical tropopause level, which may serve as an upper boundary for the LMS can be defined thermally by the
(potential) temperature lapse rate or the cold point, i.e., the local temperature minimum (Holton et al., 1995), or as an isentrope, e.g., the potential temperature level of 380 K.

Since the tropopause is strongly coupled to the temperature profile in the lower and middle atmosphere, the tropopause height is sensitive to climate changes and variability. Specifically, the tropopause height has been shown to be a robust fingerprint of
anthropogenic climate change. Increasing concentrations of well-mixed greenhouse gases (GHG) lead to a warming of the troposphere and cooling of the stratosphere, reinforced by the depletion of stratospheric ozone. Warming of the troposphere

and cooling of the stratosphere result in a tropopause rise (e.g., Seidel et al., 2001; Santer et al., 2004; Seidel and Randel, 2006). In addition to radiative effects, stratospheric dynamics, as manifested by the Brewer-Dobson circulation (BDC), affect the location of the tropopause and the equator-to-pole contrast in tropopause height (Birner, 2010b).


A rise of the global tropopause of the order of $50$–$100\,\mathrm{m/decade}$ has been reported in many studies of radiosonde (e.g., Seidel and Randel, 2006; Xian and Homeyer, 2019; Meng et al., 2021) and satellite observations (e.g., Schmidt et al., 2008; Meng et al., 2021) as well as in reanalysis data (e.g., Wilcox et al., 2012; Xian and Homeyer, 2019) and climate models (e.g., Santer, 2003a) and for different tropopause definitions. Even though pressure surfaces are rising together with the tropopause as a result of the thermal expansion of the troposphere (e.g., Eichinger and Šácha, 2020), it has been found that tropopause pressure trends are negative of the order of $-1\,\mathrm{hPa/decade}$, apparently exceeding isobaric height trends (e.g., Santer, 2003a; Seidel and Randel, 2006; Wilcox et al., 2012). Natural variability has been found to play a minor role in tropopause trends (Sausen and Santer, 2003; Meng et al., 2021). Instead, model experiments by Santer (2003b) suggest that $80\,\%$ of the tropopause rise between 1979–1999 are attributable to anthropogenic influence, primarily increasing greenhouse gas emissions and the resulting warming of the troposphere and cooling of the stratosphere. The tropospheric warming was found to cause a persistent lifting of the tropopause, even as contrasting temperature effects are expected in the stratosphere, where ozone recovery is causing warming while increasing GHG load is exerting a cooling influence (Pisoft et al., 2021; Meng et al., 2021).

In the tropics, the cold point is rising together with the lapse rate tropopause (e.g., Tegtmeier et al., 2020; Zou et al., 2023). Temperature trends in the tropical lower stratosphere and the cold point show large uncertainties. Tegtmeier et al. (2020) report small negative cold point temperature trends for the time period 1979–2005 for different reanalyses. Results by Zou et al. (2023) agree for the same time period, but suggest a warming of the tropical tropopause from 2006–2021 in ERA5. A similar trend behavior has been reported by Fu et al. (2019) for TTL temperatures from satellite observations. Before 2000, temperatures show small but significant negative trends, consistent with Tegtmeier et al. (2020), but no trend thereafter.

The largest trends in tropopause height are usually apparent in the subtropics around the tropopause break, which can be associated with a transition from lower extratropical values to significantly higher levels characteristic for the tropical tropopause. This effect can be understood as a broadening of the tropics and has been observed in radiosonde and satellite data (e.g., Seidel and Randel, 2007; Meng et al., 2021) as well as reanalysis data (e.g., Lu et al., 2009; Birner, 2010a; Wilcox et al., 2012). However, evidence for recent tropical narrowing has been reported by Zou et al. (2023). Xian and Homeyer (2019) find a dipole structure with regions of increasing and decreasing tropopause altitude around the mean tropopause break in different zonally resolved reanalyses for the time period 1981–2015. In latitude coordinates relative to the tropopause break, the dipole structure mostly disappears and tropopause trends become largest around the mean tropopause break which agrees with zonal mean analyses. Robust tropical widening has also been documented based on other metrics that, however, do not necessarily correlate well with variations in the location of the tropopause break (e.g., Staten et al., 2018; Grise et al., 2019). Turhal et al. (2024) report a tropical widening above $370\,\mathrm{K}$ and below $340\,\mathrm{K}$ but a narrowing of the tropospheric width in the region in between for the isentropic PV-gradient tropopause from reanalysis data between 1980–2017.

In addition to the general widening of the tropics, the frequency of double tropopause events, i.e. poleward excursions of the tropical above the extratropical tropopause, is found to have increased (Castanheira et al., 2009; Xian and Homeyer, 2019). This trend likely reflects an increase in baroclinicity in UTLS, driven by the GHG-induced climate change (Castanheira et al., 2009).

In addition to the observed long-term changes of the tropopause location, climate model projections agree on an acceleration of the Brewer-Dobson circulation (BDC) as a consequence of the greenhouse-gas induced climate change (e.g., Butchart, 2014). According to Oberländer-Hayn et al. (2016) and Šácha et al. (2024), it is more precise to speak about a lifting of the circulation, which is connected to the tropopause expansion itself. Stratospheric temperature changes linked to stratospheric ozone additionally influence the BDC evolution. In response to the increased tropical upwelling (extratropical downwelling), resulting in adiabatic cooling (heating), the temperature structure of the stratosphere is expected to change.

Observations of the BDC behavior are only possible indirectly and the different approaches give a less clear picture about recent BDC trends than model simulations. Thompson and Solomon (2009) for example, report trends of lower stratospheric temperature and ozone between 1979–2006 from satellite observations that agree with an increased BDC, consistent with findings by Tegtmeier et al. (2020) from reanalysis data and observations for the same time period (1979-2005). Fu et al. (2019) use satellite observations of stratospheric temperatures to estimate an acceleration of the mean BDC by $1.7\,\%/\mathrm{decade}$ between 1980–2018 but a recent deceleration between 2000–2018. This is consistent with tropical lower stratospheric temperature trends close to zero within this period, inferred by Zou et al. (2023) from reanalyses data. The temperature reduction in the tropical tropopause region and at the cold point reported for the time period 1979–2005 by Tegtmeier et al. (2020) is consistent with increased tropical upwelling. However, different reanalyses often show a significant spread when compared, whether in terms of, e.g., temperature trends in the TTL region (e.g., Tegtmeier et al., 2020) or dynamical tropical upwelling (e.g., Šácha et al., 2024).

All in all, the thermodynamic structure of the lowermost stratosphere can be expected to change in response to increasing greenhouse gas concentrations and the recovery of stratospheric ozone (e.g. IPCC, 2023, Charlesworth et al., 2023). This study investigates the effect of long-term changes of the tropopause and the lower stratospheric potential temperature structure and their impact on the mass of the lowermost stratosphere on the basis of reanalysis data for the time period 1979–2019. We compare results from five modern reanalyses: ERA5, ERA-Interim, MERRA-2, JRA-55 and JRA3Q. To illustrate our metrics for examining the LMS structure, we present results for ERA5 before generalizing our findings for all data sets. We focus on the time period after 1998, which marks the anticipated beginning of stratospheric ozone recovery.

The data and regression model used for this study are described in Sect. 2. Thereafter, Sect. 3 presents and discusses the results, divided into subsections concerning the temporal evolution of the tropopause (Sect. 3.1) and the lower stratospheric potential temperature structure (Sect. 3.2) as well as the mass of the lowermost stratosphere (Sect. 3.3). Sect. 3.3 is further divided into a consideration of the LMS boundary surfaces, an LMS climatology and LMS mass trends between 1998–2019. The final conclusions are presented in Sect. 4.

## 2 Data and methods

### 2.1 Data

Reanalysis data sets are valuable assets to study atmospheric features from the process to quasi-climatological scales. In the
UTLS we know that differences between the various reanalysis exist as well as differences to observations (Fujiwara et al.,
2022). We therefore conduct our analysis for five reanalyses, four of which have been widely used in recent years, i.e. ERA5
(Hersbach et al., 2020), ERA-Interim (Dee et al., 2011), MERRA-2 (Gelaro et al., 2017) and JRA-55 (Kobayashi et al., 2015)
as well as the recently published JRA3Q (Kosaka et al., 2024). For the introduction and illustration of our metrics to assess the
structure of the LMS, we use ERA5 before we generalize our findings to all other data sets.

We use ERA5 monthly mean data for the time period 1979–2019 (Hersbach et al., 2020), 1979 marking the beginning of
the satellite era. For the time period 2000–2006, the sub-reanalysis ERA5.1 replaces ERA5, correcting the reanalysis for a
cold bias in the lower stratosphere (Simmons et al., 2020). We calculated the monthly mean data and the associated standard
deviations from the hourly available ERA5 data. ERA5 has 137 hybrid sigma-pressure model levels in the vertical. ERA5 thus
has the finest vertical resolution among modern reanalyses. In addition to the model level data, ECMWF provides the model
data interpolated onto 37 pressure levels between $1000\,\mathrm{hPa}$ and $1\,\mathrm{hPa}$. On pressure levels, the vertical grid spacing of ERA5
in the UTLS is $25\,\mathrm{hPa}$ to $50\,\mathrm{hPa}$, which corresponds to roughly 600–1200 m. Furthermore, model data interpolated on the
$2\,\mathrm{PVU}$ surface is available from ECMWF directly (Hersbach et al., 2020). We use ERA5 data on pressure levels and $2\,\mathrm{PVU}$
on a regular horizontal grid with a constant grid spacing of $0.25°$ in longitude and latitude.

In addition, we use monthly mean data from ERA-Interim (Dee et al., 2011), MERRA-2 (Gelaro et al., 2017), JRA-55
(Kobayashi et al., 2015) and JRA3Q (Kosaka et al., 2024) for the same time period as ERA5. However, ERA-Interim is
only available until 2018 and the MERRA-2 time series begins in 1980. We use all data on the provided pressure levels and
regular horizontal grids. ERA-Interim provides 23 pressure levels, with the same spacing as ERA5 in the UTLS, interpolated
from 60 hybrid sigma–pressure model levels and a horizontal grid spacing of $0.75°$ latitude x $0.75°$ longitude. MERRA-2
comes with 42 pressure levels from 72 model levels and a horizontal grid of $0.5°$ x $0.625°$. JRA-55 contains 37 pressure levels
from 60 model levels and a horizontal grid of $1.25°$ x $1.25°$. JRA3Q provides 40 pressure levels from 100 model levels and
a horizontal grid of $1.25°$ x $1.25°$. Tab.1 presents an overview of the vertical pressure levels available in the UTLS across the
different reanalyses. A more detailed comparison of the vertical levels across the reanalysis can be found in Fujiwara et al.
(2022) and JMA (2022).

| Data set | UTLS pressure levels [hPa] | | | | | | | | | | | | | | |
|---|---|---|---|---|---|---|---|---|---|---|---|---|---|---|---|
| ERA5 | 50 | 70 | | 100 | 125 | 150 | 175 | 200 | 225 | 250 | 300 | 350 | 400 | 450 | 500 |
| ERA-Interim | 50 | 70 | | 100 | 125 | 150 | 175 | 200 | 225 | 250 | 300 | 350 | 400 | 450 | 500 |
| MERRA-2 | 50 | 70 | | 100 | | 150 | | 200 | | 250 | 300 | 350 | 400 | 450 | 500 |
| JRA-55 | 50 | 70 | | 100 | 125 | 150 | 175 | 200 | 225 | 250 | 300 | 350 | 400 | 450 | 500 |
| JRA3Q | 50 | 70 | 85 | 100 | | 150 | 175 | 200 | 225 | | | 350 | 400 | | 500 |

**Table 1.** Overview of the vertical pressure levels available in the UTLS across the reanalyses used in this study.

## 2.2 Tropopause detection

To determine the lapse rate tropopause according to WMO (1957) from all five reanalysis data sets, we apply an algorithm closely following that of Birner (2010a), based on the work of Reichler et al. (2003). The algorithm computes the temperature lapse rate on $p^\kappa = p^{R_d/c_p}$ half levels (where $p$ is the pressure, $R_d$ is the specific gas constant and $c_p$ is the heat capacity at constant pressure for dry air), according to equations 1)–4) in Reichler et al. (2003). Subsequently, the algorithm identifies the lowest half level at which the lapse rate becomes smaller than $2\,\mathrm{K/km}$. Following Birner (2010a), a preliminary tropopause is then identified by linear interpolation of $p^\kappa$ and all other variables to a lapse rate of exactly $2\,\mathrm{K/km}$ (stratification threshold). Second, the algorithm checks whether the lapse rate remains on average below $2\,\mathrm{K/km}$ for all higher levels within $2\,\mathrm{km}$ (thickness criterion). Therefore, a temporal level at $2\,\mathrm{km}$ distance to the preliminary tropopause is added and the algorithm successively checks the average lapse rate for all levels between the preliminary tropopause and $2\,\mathrm{km}$ distance. If the preliminary tropopause does not fulfill the thickness criterion, the next higher half level fulfilling the stratification threshold is tested. The search range is limited to 500–75 hPa to avoid unreasonable values, for example due to surface inversions. At high latitudes, especially over the Antarctic continent in austral winter, it can be difficult to obtain a meaningful lapse rate tropopause due to weak stratification (Zängl and Hoinka, 2001). We therefore limit the tropopause pressure at latitudes $>50°$ to a minimum pressure of $150\,\mathrm{hPa}$. If the detected tropopause pressure falls below this limit, the thickness criterion is suspended and the highest pressure ($<500\,\mathrm{hPa}$) corresponding to a lapse rate of $2\,\mathrm{K/km}$ represents the tropopause. Missing values are filled by bilinear interpolation.

In addition to the thermal tropopause, we present an analysis of the dynamical tropopause in ERA5. ECMWF provides ERA5 model level data interpolated onto the $2\,\mathrm{PVU}$ isosurface which serves as the dynamical tropopause for this study. The $89\,\mathrm{hPa}$ isobar is taken as tropopause cap for cases when $2\,\mathrm{PVU}$ is at lower pressure than $89\,\mathrm{hPa}$ to avoid unreasonably large altitudes for the tropopause towards the equator (Hersbach et al., 2020; ECMWF, 2016). Not all reanalyses provide data on the $2\,\mathrm{PVU}$ level or a potential vorticity variable in general. As consistency of the dynamical tropopause across the reanalyses is therefore not guaranteed, we present trends regarding the $2\,\mathrm{PVU}$ tropopause only for ERA5 as an example.

Between 20°N–20°S, the cold point serves as an alternative metric to estimate the tropical tropopause (Holton et al., 1995).

To deduce the cold point tropopause from all five reanalyses, we determine pressure and potential temperature at the level corresponding to a lapse rate of $0\,\mathrm{K/km}$.

## 2.3 LMS mass

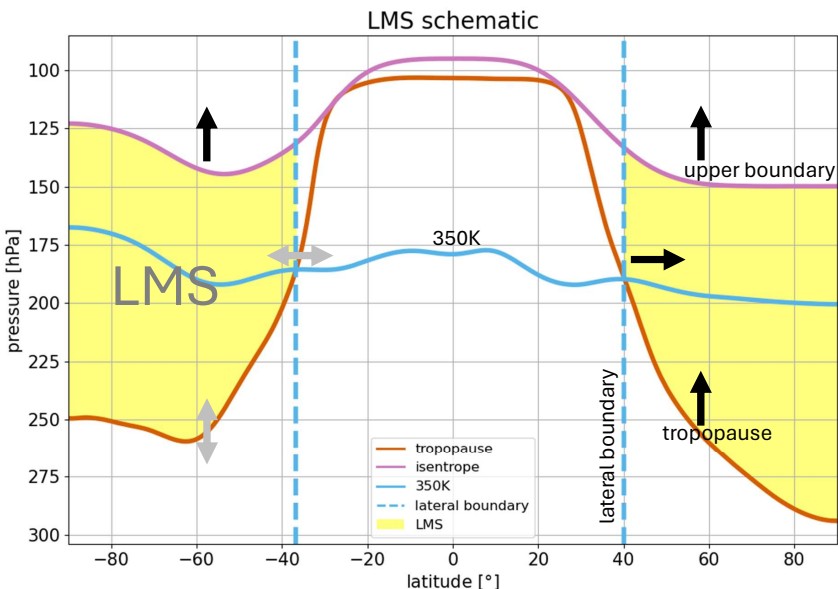

**Figure 1.** Schematic of the LMS. The LMS mass is enclosed by a lower boundary (tropopause, red), an upper boundary (purple) and a lateral boundary (blue vertical lines). The lateral boundary in this study is defined as the intersection between the tropopause and the $350\,\mathrm{K}$ isentrope, approximating the location of the subtropical jet streams and the maximum PV-gradient, marking a transport barrier. The arrows indicate the long term changes of the boundary surfaces, affecting the LMS mass as presented in Sect. 3.1 and 3.3.1. Figure 8 shows the location of the different LMS boundary surfaces compared in this study.

In order to determine the LMS mass, appropriate LMS boundary surfaces have to be defined. The tropopause serves as the lower boundary. For the upper LMS boundary, unlike most approaches, we will not only use a fixed isentropic value. Instead, in this study the upper LMS boundary surface depends on the potential temperature at the tropical tropopause. Therefore, we determine the pressure at the isentrope corresponding to the mean potential temperature at the tropical ($10°$N–$10°$S) lapse rate tropopause (PPT10mean) as well as at the cold point (PPTcp10mean) for every time step. In many LMS studies, the $380\,\mathrm{K}$

isentrope has been used to approximate the upper LMS edge (e.g., Appenzeller et al., 1996; Wang and Fu, 2021). In this study, we compare $380\,\mathrm{K}$ with the dynamically defined upper boundaries (PPT10mean and PPTcp10mean).

The lateral LMS boundary in this study is defined as the latitude at which the respective monthly mean tropopause intersects with the $350\,\mathrm{K}$ isentrope. This intersection is determined by the sign change of the pressure difference between the tropopause and the isentrope. The latitude of intersection between the tropopause and $350\,\mathrm{K}$ roughly marks the location of the subtropical

jet stream (e.g., Manney et al., 2011; Gettelman et al., 2011). Specifically, at $350\,\mathrm{K}$, the lapse rate tropopause and the $2\,\mathrm{PVU}$ surface are near the isentropic PV-gradient tropopause, a transport barrier separating the extratropical lower stratosphere from the tropical troposphere (Kunz et al., 2011; Turhal et al., 2024). The isentropic PV-gradient tropopause for the time period 1979–2019 was derived according to Turhal et al. (2024) and interpolated to pressure and potential temperature fields between $50°\mathrm{N}$–$50°\mathrm{S}$ to be directly comparable to the lapse rate tropopause and $2\,\mathrm{PVU}$.


Following Appenzeller et al. (1996), the LMS mass $M(t)$ is obtained from the three dimensional integral of the pressure differences between the tropopause $p_1(\lambda, \Phi, t)$ and the pressure at the upper boundary $p_2(\lambda, \Phi, t)$ at every grid point and time step:

$$M(t) = \int\limits_{0}^{2\pi} \int\limits_{\Phi_1(\lambda, t)}^{\Phi_2(\lambda, t)} \int\limits_{p_1(\lambda, \Phi, t)}^{p_2(\lambda, \Phi, t)} -\frac{1}{g} dp \cos \Phi d\Phi d\lambda, \tag{1}$$

$g$ is the gravity constant, $\lambda$ is the longitude, $\Phi$ is the latitude and $dp$ is the pressure difference. $\Phi_1(\lambda, t)$ and $\Phi_2(\lambda, t)$ denote the lateral boundaries, defined by the intersection between the tropopause and the $350\,\mathrm{K}$ isentrope for every longitude and time step. All five reanalyses are used to calculate the LMS mass with respect to the lapse rate tropopause as the lower LMS boundary. For ERA5, the $2\,\mathrm{PVU}$ dynamical tropopause is used in addition.

## 2.4 Regression tools

We use the dynamic linear regression model (DLM) by Laine et al. (2014) for the time series analyses in this study. The DLM has proved to be useful for trend estimation in many studies (Laine et al., 2014; Ball et al., 2017, 2018, 2019; Karagodin-Doyennel et al., 2022; Bognar et al., 2022; Minganti et al., 2022) due to its ability to identify non-linear trends, smoothly varying in time without prior specification of possible inflection dates. Another strength of DLM is the rigorous treatment of uncertainties in the data and the regression coefficients by simultaneously estimating all model components. These model com-

ponents include a non-linear background trend, a seasonal cycle composed of a 6 and a 12 months period and an autoregressive process. The user is able to add regressor variables, assessing causes of natural variability. For the DLM trend analyses in this study, we use regressors to account for El-Niño/Southern Oscillation (NOAA), the quasi-biennial oscillation at 30 and $50\,\mathrm{hPa}$ (FU-Berlin) as well as for stratospheric (volcanic) aerosol optical depth (SAOD) (Thomason et al., 2018). The same regressors have been used in different studies investigating changes of UTLS characteristics (e.g., Seidel and Randel, 2006; Meng et al.,

2021; Tegtmeier et al., 2020; Zou et al., 2023). All regressors are normalized and, except SAOD, centered around zero. In order to assess the robustness of the trends and the effect of the regressors, we conducted sensitivity tests in which the DLM was run with and without regressors. Overall, the results showed no strong dependency on the used regressors. However, tropical tropopause pressure trends, for example, become more significant when regressors are used (Figs. 2, A2).

The DLM is a Bayesian model based on the principles of Kalman filtering. The algorithm estimates the future system state based on the previous estimates and the variance of the underlying data. The DLM uses an iterative process based on the

Markov chain Monte Carlo (MCMC) method to infer possible system state time series from the Kalman algorithm outputs. Further details about the DLM can be found in Laine et al. (2014). The python based model code (dlmmc) is publicly available (Alsing, 2019).

In this study, the DLM runs provide 2000 possible model state estimates after an additional 1000 samples that are considered as warm-up and discarded. Sensitivity experiments conducted as part of this study have shown that increasing the number of DLM samples (to e.g. 10 000) does not significantly improve the results, but comes at a considerable computational cost (see for example Fig. A1).

The focus is on the hidden mean trend estimates. In the following, the DLM mean trend state time series is referred to as DLM
trend state. The terms "trend" or "change" denote the DLM trend state difference between two dates, usually between January 1998 and December 2019. The fraction of DLM samples resulting in a negative (positive) change between two dates serves as evaluation of the confidence of the trend. The use of the term "statistical significance" refers to the probability of an overall decrease (increase) >95 % instead of frequentist significance tests. Ball et al. (2018, 2019) use this approach to evaluate their ozone trend analysis.

## 3.1 Temporal evolution of the tropopause

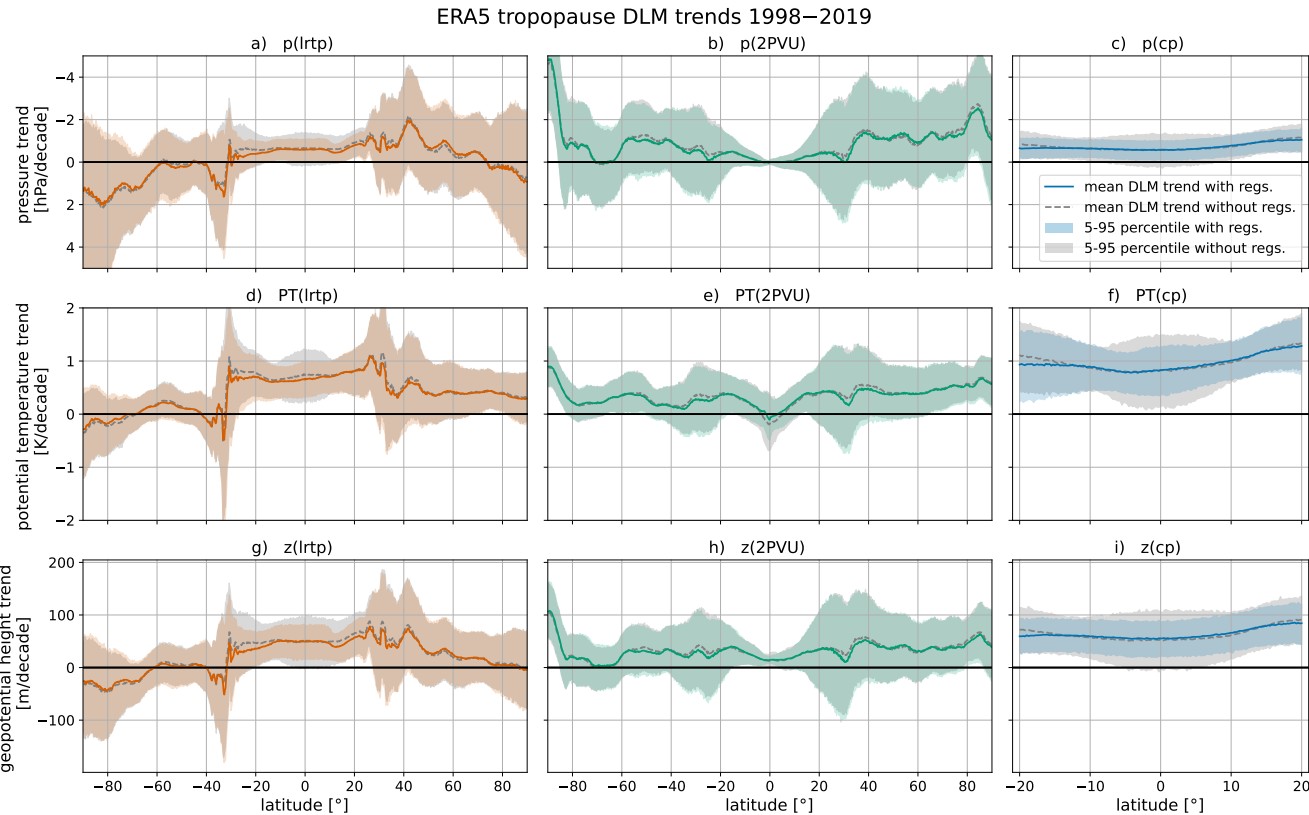

**Figure 2.** Zonal mean tropopause pressure (upper panels), potential temperature (middle row) and geopotential height (lower panels) trends between 1998–2019 as estimated by the DLM from ERA5 data. Note that the pressure trend axis is inverted because decreasing pressure is associated with an upward trend in the pressure coordinate system. The lines show the mean DLM trends, the shading denotes the 5–95 percentile. DLM results including regressors are presented as solid lines and in color, DLM results without regressors are shown as dashed lines and in grey. Red color is associated with the lapse rate tropopause (left panels), green color (center column) indicates the 2 PVU dynamic tropopause (capped at 89 hPa in the tropics) and blue color (right panels) the cold point.

Figure 2 shows the DLM trend results over the period 1998–2019 for the zonal mean tropopause pressure (upper panels), potential temperature (middle row) and geopotential height (lower panels) in ERA5. In the NH mid-latitudes, both the ERA5 lapse rate tropopause and the 2 PVU dynamical tropopause show negative pressure trends of around $-1\,\mathrm{hPa/decade}$ (Fig. 2a and b). This pressure decrease corresponds to a lifting of the tropopause (Fig. 2g and h) and is consistent with the positive potential temperature trends (Fig. 2d and e). ERA-Interim, MERRA-2, JRA3Q and JRA-55 exhibit remarkably similar lapse rate tropopause trends, which enhances the robustness of the findings (Fig. 3a and c). Furthermore, similar tropopause trends have

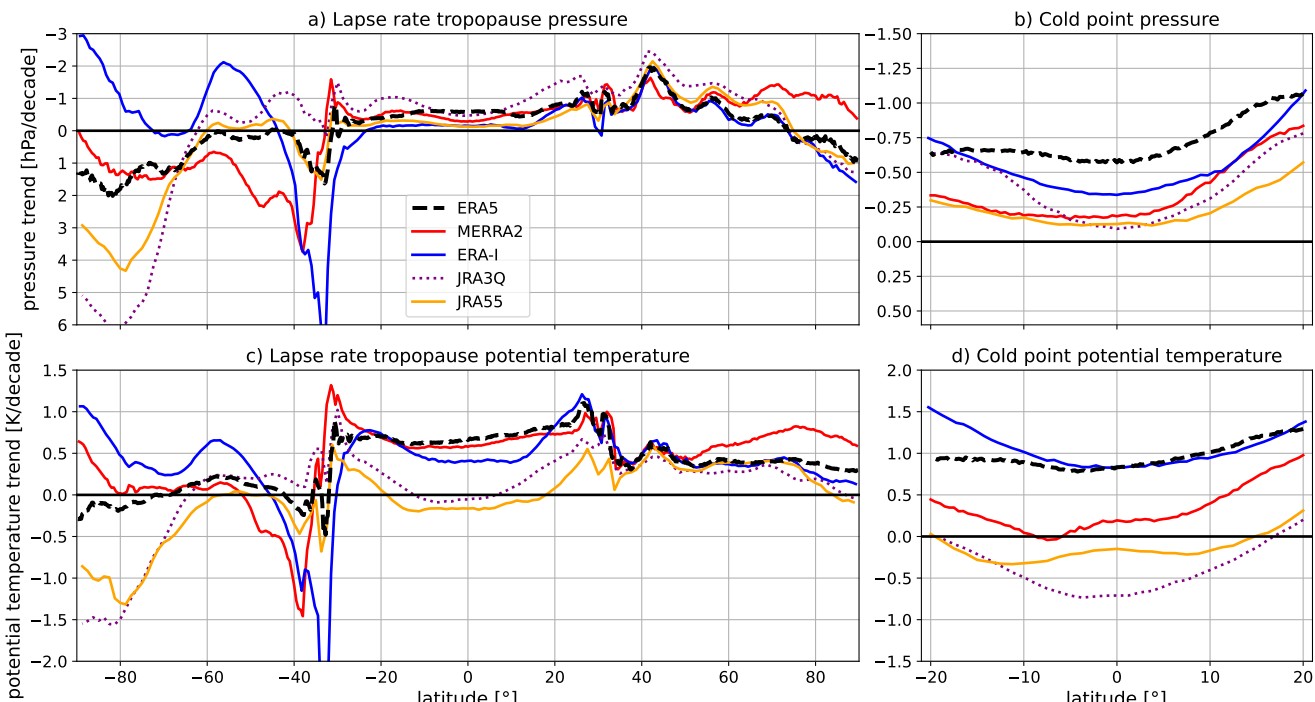

**Figure 3.** Zonal mean lapse rate tropopause (left panels) and cold point (right panels) trends between 1998–2019 as estimated by the DLM from different reanalyses. The reanalyses compared are ERA5 (black, dashed line), MERRA-2 (red, solid), ERA-Interim (blue, solid), JRA3Q (purple, dotted) and JRA-55 (orange, solid). The upper panels displays pressure trends, the lower panels show potential temperature trends. Note that the pressure trend axis is inverted because decreasing pressure is associated with an upward trend in the pressure coordinate system. The lines show the mean DLM trends. For the sake of clarity, the presentation of uncertainty has been omitted but is of the same order of magnitude as in Fig. 2a, c, d and f.

been reported by Wilcox et al. (2012) for the time period 1989–2007 and Meng et al. (2021) for the time period 1980–2020. The tropical and subtropical lapse rate tropopause in ERA5 exhibits statistically significant upward trends accompanied by

negative trends in pressure (ca. $-0.5\,\mathrm{hPa/decade}$, Fig. 2a and g) and more pronounced potential temperature trends compared to the extratropics (ca. $0.7\,\mathrm{K/decade}$, Fig. 2d). Since the $2\,\mathrm{PVU}$ tropopause is capped at a fixed pressure of $89\,\mathrm{hPa}$ in the tropics, the respective pressure trends in the inner tropics are not meaningful (Fig. 2b, e and h).

As Figs. 2c and i reveal, the zonal mean cold point in ERA5 is rising, accompanied by a statistically significant pressure decrease of on average $-0.7\,\mathrm{hPa/decade}$ (Fig. 2c), which is in agreement with findings of Tegtmeier et al. (2020). The upward

tendency of the cold point in ERA5 is joined by a potential temperature increase of on average $1\,\mathrm{K/decade}$ (Fig. 2f).

All considered reanalyses agree on negative pressure trends at the tropical lapse rate tropopause and the cold point (Fig. 3a and b), whereas potential temperature trends differ in sign (Fig. 3c and d). Specifically, while ERA5, ERA-Interim and MERRA-2

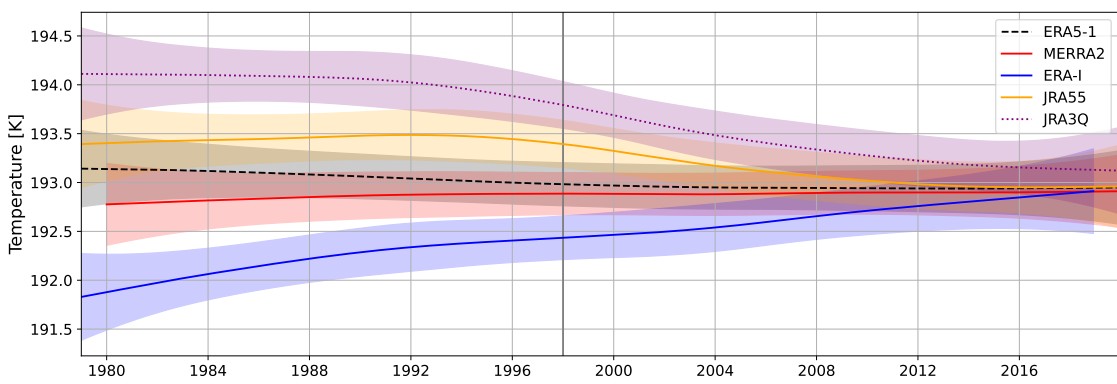

**Figure 4.** DLM trend state for the average tropical ($10°$N-$10°$S) temperature at $100\,\mathrm{hPa}$ in ERA5 (black, dashed line), MERRA-2 (red, solid), ERA-Interim (blue, solid), JRA-55 (orange, solid) and JRA3Q (purple, dotted) for the time period 1979–2019. Shading denotes the range of 2 standard deviations. The ERA-Interim time series ends in 2018, MERRA-2 begins in 1980.

agree on positive potential temperature trends between $10°$N–$10°$S, JRA3Q and JRA-55 suggest the opposite. These discrepancies can be attributed to contrasting absolute temperature trends in the TTL region suggested by the reanalyses, such as those
visible at the $100\,\mathrm{hPa}$ pressure level (Fig. 4).

SH tropopause trends are less conclusive than in the NH. Within ERA5, lapse rate and $2\,\mathrm{PVU}$ dynamical tropopause pressure trends differ in sign at most latitudes (Fig. 2a and b). The ERA5 SH mid-latitude lapse rate tropopause generally shows positive pressure trends ranging between 0 and $+2\,\mathrm{hPa/decade}$, corresponding to a sinking of the tropopause (Fig. 2a and g) while
$2\,\mathrm{PVU}$ pressure trends in this region amount to around $-0.8\,\mathrm{hPa/decade}$ (Fig. 2b). Throughout the reanalyses, lapse rate tropopause trends show a large spread in the SH extratropics (Fig. 3a and c). Like in ERA5, a positive pressure trend of the SH mid-latitude lapse rate tropopause between $30°$–$40°$S is also evident in ERA-Interim, MERRA-2 and JRA-55 but not in JRA3Q (Fig. 3a). A similar tropopause behavior has been observed by Xian and Homeyer (2019) for the reanalyses JRA-55, MERRA-2 and CFSR. On the other hand, Xian and Homeyer (2019) do not observe such a downward trend of the SH mid-
latitude lapse rate tropopause in ERA-Interim and it seems to oppose the findings by Wilcox et al. (2012) and Meng et al. (2021). However, it has to be noted that in this study, we focus on tropopause changes after 1998. The non-linear DLM trend analysis for the entire time series 1979–2019 reveals a trend reversal of the ERA5 lapse rate tropopause pressure around the year 2000. This trend reversal is evidenced by the fact that the DLM pressure trend state reaches its minimum around the year 2000 (Fig. 5). Accordingly, the DLM results suggest decreasing pressure of the ERA5 SH mid-latitude lapse rate tropopause
before the year 2000 and increaing pressure thereafter. The DLM trend state of $2\,\mathrm{PVU}$ and cold point pressure can be found in the appendix (Fig. A3).
In the polar regions of both hemispheres, ERA5 suggests increasing tropopause pressure, which is evident in most reanalyses

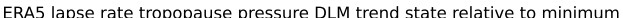

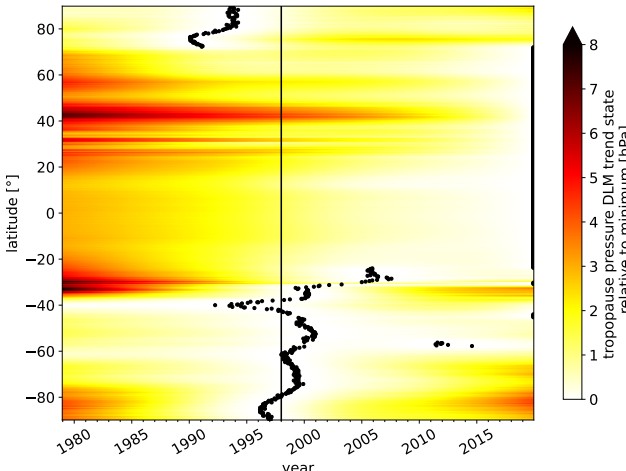

**Figure 5.** DLM trend state time series of pressure at the lapse rate tropopause in ERA5 for the time period 1979–2019. The panels show the pressure trend state time series for all latitudes (y-axis). The minimum pressure trend state for every grid point is highlighted by the black dots and can indicate a trend reversal from decreasing to increasing pressure. The trend state time series are presented relative to the respective minimum but in absolute numbers [hPa] (color shading). The black vertical line marks the beginning of the year 1998.

and highly significant in the SH in JRA3Q and JRA-55 (Figs. 2a and 3a). The positive pressure trends at the lapse rate tropopause above high latitudes could be the result of polar ozone recovery and associated increasing temperatures in the polar lower stratosphere (LS) after 1998. This is consistent with the minimum lapse rate tropopause pressure DLM trend state in the polar regions, estimated around the time of ozone trend reversal (around the year 2000) (Fig. 5). A descent of the SH polar lapse rate tropopause has also been observed by Wilcox et al. (2012) and in some of the reanalyses analyzed by Xian and Homeyer (2019). Moreover, the downward trend of the lapse rate tropopause at high and mid-latitudes could be linked to an acceleration of the BDC (Birner, 2010b). Indeed, age of air and long-lived trace gas measurements indicate an accelerating BDC in the SH after about the year 2000 (Strahan et al., 2020; Ploeger and Garny, 2022).

### 3.2 Temporal evolution of the lower stratospheric potential temperature structure

In addition to trends of the tropopause location, the location of isentropic surfaces is an important measure for structural changes of the UTLS region. Using ERA5 as an example, Fig. 6a–c show the pressure trend on isentropes at $380\,\mathrm{K}$, $400\,\mathrm{K}$ and $430\,\mathrm{K}$ for the time period 1998–2019 as estimated by the DLM. The average pressure trends range between $-0.3$ to $+0.6\,\mathrm{hPa/decade}$, which corresponds to an absolute pressure change of up to $1\,\mathrm{hPa}$ during the 21 year period, albeit exhibiting large uncertainties, especially in the extratropics. The pressure trends between 380–430 K are positive in the extratropics. In the tropics and subtropics, on the other hand, the pressure for the isentropes above $380\,\mathrm{K}$ is found to be decreasing, which is accompanied by a rise of the respective potential temperature iso-surfaces in geopotential height (Fig. A4a). The isentropic

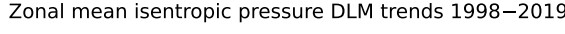

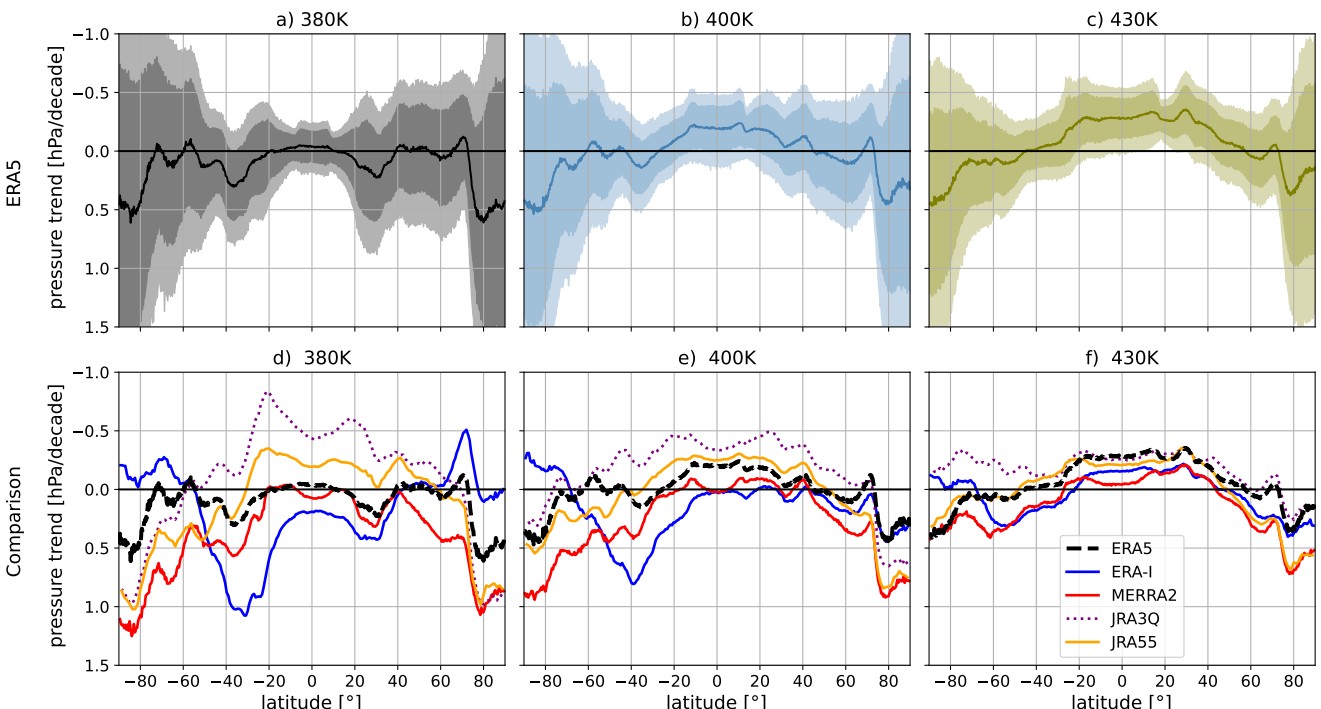

**Figure 6.** Zonal mean pressure trends at isentropes $380\,\mathrm{K}$ (left panels), $400\,\mathrm{K}$ (middle column) $430\,\mathrm{K}$ (right panels) as estimated by the DLM for the time period 1998–2019. As an example, ERA5 isentropic pressure trends are depicted in the upper panels, showing the mean trend (solid line) together with the associated standard deviation (dark shading) and the 5–95 percentile. The lower panels compare results for the different reanalyses, i.e. ERA5 (black, dashed line), MERRA-2 (red, solid), ERA-Interim (blue, solid), JRA3Q (purple, dotted) and JRA-55 (orange, solid). For the sake of clarity, the presentation of uncertainty has been omitted in the lower panels. Note that the pressure trend axis is inverted because decreasing pressure is associated with an upward trend in the pressure coordinate system. The trends are given in hPa/decade.

pressure trends are hardly statistically significant but robust within the reanalyses (except ERA-Interim) (Fig. 6d–f). However,

while the $380\,\mathrm{K}$ isentrope in ERA5 and MERRA-2 shows no pressure trend in the tropics and subtropics, JRA-55 and JRA3Q show a significant pressure decrease (Fig. 6d). Decreasing pressure of lower stratospheric isentropes in the tropics and increasing pressure in the extratropics is consistent with a strengthened BDC, which is associated with adiabatic cooling in the tropical LS and warming in the extratropical LS.

The magnitude of the isentropic pressure trend for a given latitude bin varies at different potential temperatures. In the tropics

and subtropics in ERA5, higher isentropes are rising at a faster rate than lower ones (Fig. 6a–c). This indicates a weakening of the potential temperature gradient with pressure, and accordingly a change of the stratification in the lower stratosphere. Such lower stratospheric isentropic pressure trends could potentially be linked to TIL changes (e.g., Gettelman and Wang, 2015;

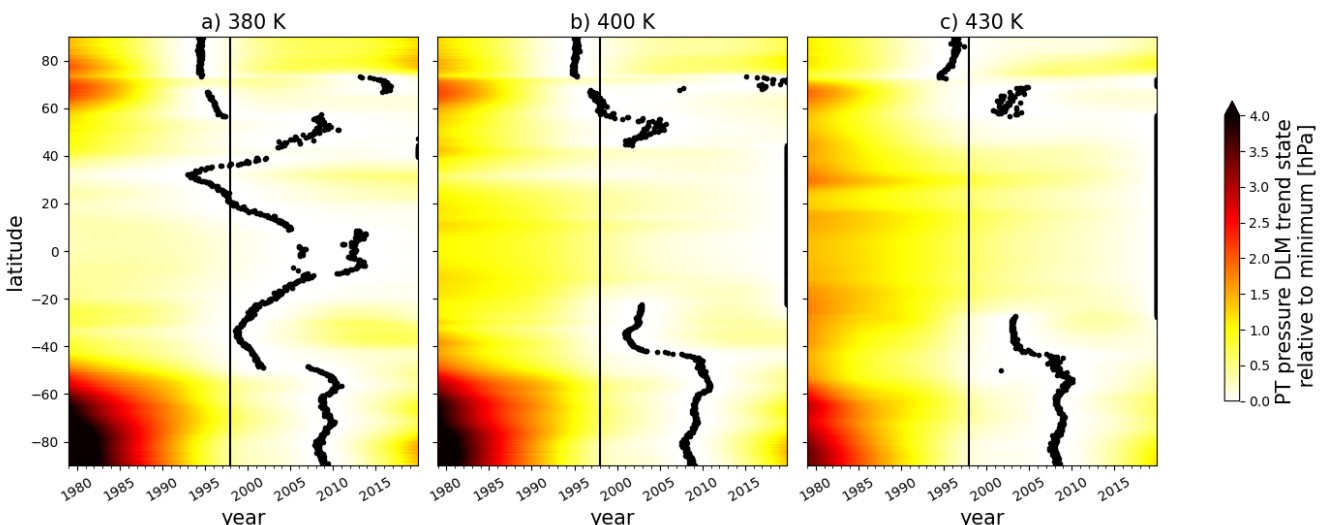

**Figure 7.** Isentropic pressure DLM trend state time series for the time period 1979–2019 at 380 K (a), 400 K (b) and 430 K (c) in ERA5 for all latitudes (y-axis). The minimum pressure trend state for every grid point is highlighted by the black dots and can indicate a trend reversal from decreasing to increasing pressure. The pressure trend state time series are presented relative to the respective minimum but in absolute numbers [hPa] (color shading). The black vertical line marks the beginning of the year 1998.

Boljka and Birner, 2022), at least in the tropics. However, this behavior is only evident in the ECMWF reanalyses (Fig. 6d–f). While the average pressure changes discussed in this paragraph provide valuable insights, examining the full, non-linear DLM trend state time series can offer a more comprehensive view of the physical mechanisms driving trend evolution.

In addition to the average isentropic pressure trends between 1998–2019, it is worth considering the entire DLM trend state time series, as the DLM is able to identify nonlinear trends. Taking ERA5 as an example, the DLM estimates a trend reversal from decreasing to increasing pressure at isentropes between 380–430 K in the extratropics, spread around the year 2000. This trend reversal is evident from the minimum in the DLM trend state (Fig. 7). At SH high latitudes, isentropes reach their minimum pressure trend state later, between 2006–2008, but pressure changes are most pronounced in this region. Especially in SH high latitudes, decreasing isentropic pressure between the 1980s and the 2000s is consistent with a cooling of the lower stratosphere due to ozone decline. This relationship between lower stratospheric temperatures and isentropic pressure directly follows from the definition of potential temperature. Ozone recovery leads to a warming of the stratosphere, consistent with increasing isentropic pressure in the lower stratosphere since the 2000s. This is in good agreement with the SH stratospheric temperature studies by Fu et al. (2019). In the SH high latitudes this (potential) temperature increase is associated with a weakening of the SH polar vortex (Zambri et al., 2021), influencing polar winter ozone destruction and mixing of polar and mid-latitude air masses.

In the tropics and subtropics, continuous pressure decline accompanied by a rise of ERA5 isentropes above 380 K is evident (Fig. 7). This is consistent with increased upwelling in the context of an enhanced BDC, associated with adiabatic cooling. Furthermore, diabatic cooling of the tropical lower stratosphere can be expected from increasing GHG load in the atmosphere and a continuous decline of lower stratospheric ozone (e.g., Ramaswamy et al., 2001; Vallis et al., 2015). Such a cooling of the tropical lower stratosphere has been identified in observations (e.g., Scherllin-Pirscher et al., 2021) and reanalysis data (e.g., Tegtmeier et al., 2020) and is consistent with our analysis.

### 3.3 The mass of the lowermost stratosphere

#### 3.3.1 LMS boundary surfaces

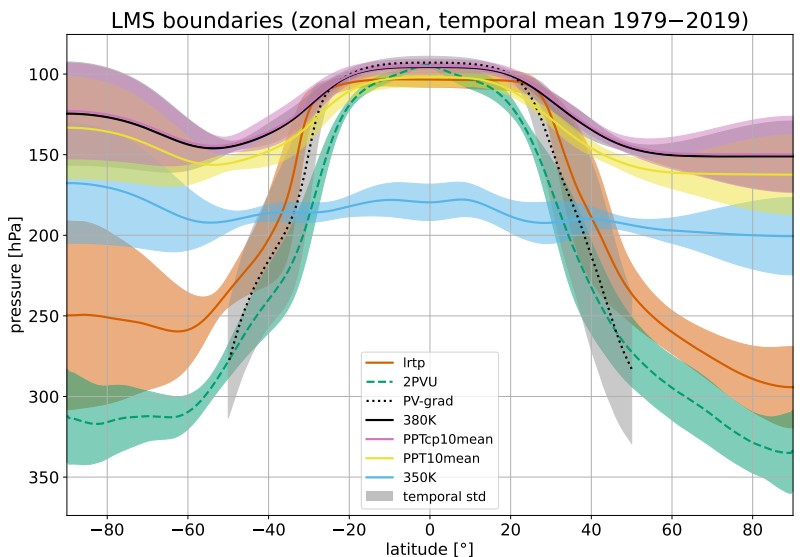

**Figure 8.** Zonal mean, temporal mean LMS boundaries between 1979–2019 using ERA5 as an example. The lines denote the mean location of the respective surfaces, the shading indicates the temporal standard deviation. The lower LMS boundary is defined by the lapse rate tropopause (red) or the 2 PVU dynamic tropopause (green, dashed line). The 380 K isentrope (black) can serve as the upper boundary. It is also possible to define dynamic upper boundaries depending on the potential temperature at the tropical tropopause. Here, the isentropes corresponding to the mean potential temperature between 10°N–10°S at the lapse rate tropopause (PPT10mean, yellow) and the cold point (PPTcp10mean, purple) are shown. The lateral LMS boundary can be approximated by the intersection between the 350 K isentrope (light blue) and the respective tropopause. This intersection is close to the isentropic PV-gradient tropopause (black, dotted line) at 350 K, which represents a transport barrier and serves as an approximation for the position of the subtropical jet stream.

The tropopause marks the lower boundary of the lowermost stratosphere (LMS). For the upper boundary, many studies choose the 380 K isentrope (e.g., Appenzeller et al., 1996; Wang and Fu, 2021). Wang et al. (2022) point out the importance

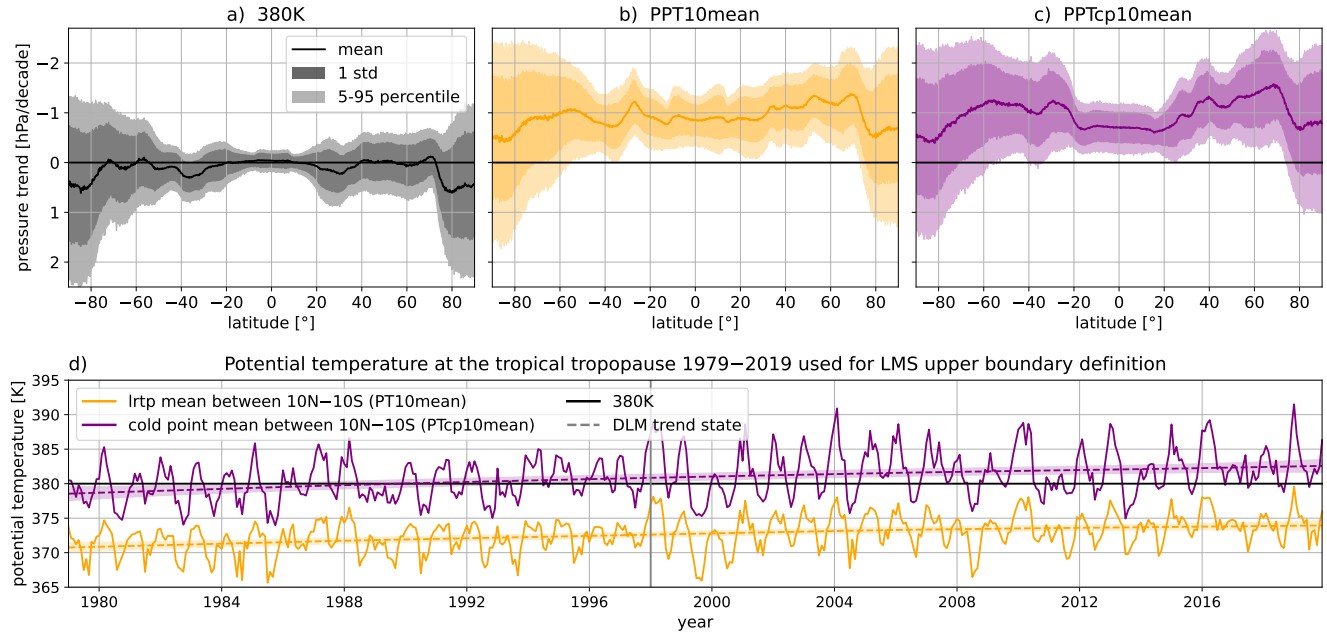

**Figure 9.** Zonal mean pressure trends of the LMS upper boundaries in ERA5 as estimated by the DLM for the time period 1998–2019 (upper panels) together with the potential temperature evolution at the tropical tropopause (lower panel). The potential temperature at the tropical (10°N–10°S) lapse rate tropopause (PT10mean, orange) and the cold point (PTcp10mean, purple) define the isentropes that serve as dynamical upper LMS boundaries (PPT10mean and PPTcp10mean). The upper LMS boundary can also be approximated with the 380 K isentrope, which is presented in addition (black). In the upper panels, the pressure trends of the respective boundaries are presented, showing the the mean trend (solid line) together with the associated standard deviation (dark shading) and the 5–95 percentile (light shading). Note that the pressure trend axis is inverted because decreasing pressure is associated with an upward trend in the pressure coordinate system. The lower panel shows the potential temperature time series (solid line) between 1979–2019 together with the DLM trend state time series (dashed lines) and the range of two standard deviations (shading). The grey vertical line marks the beginning of the year 1998.

to consider long term changes of the tropical tropopause location in different climates for LMS mass budget analyses. Their approach is to fit an isentrope between 360–390 K to the tropical lapse rate tropopause. Similar to Wang et al. (2022), we define an upper LMS boundary with respect to the tropical lapse rate tropopause that allows for continuous variations of the boundary surface. To do so, we determine the mean isentropic pressure corresponding to the potential temperature at the tropical (10°N–10°S) lapse rate tropopause (PPT10mean). For comparison, we determine a second upper boundary surface based on the potential temperature at the cold point (PPTcp10mean) in the same way. The location of the different boundary surfaces used for LMS definition are displayed in Fig. 8 using ERA5 as an example.

As evident from Fig. 9 and 10, a fixed upper LMS boundary (e.g., 380 K) cannot capture the variability and long-term evolution

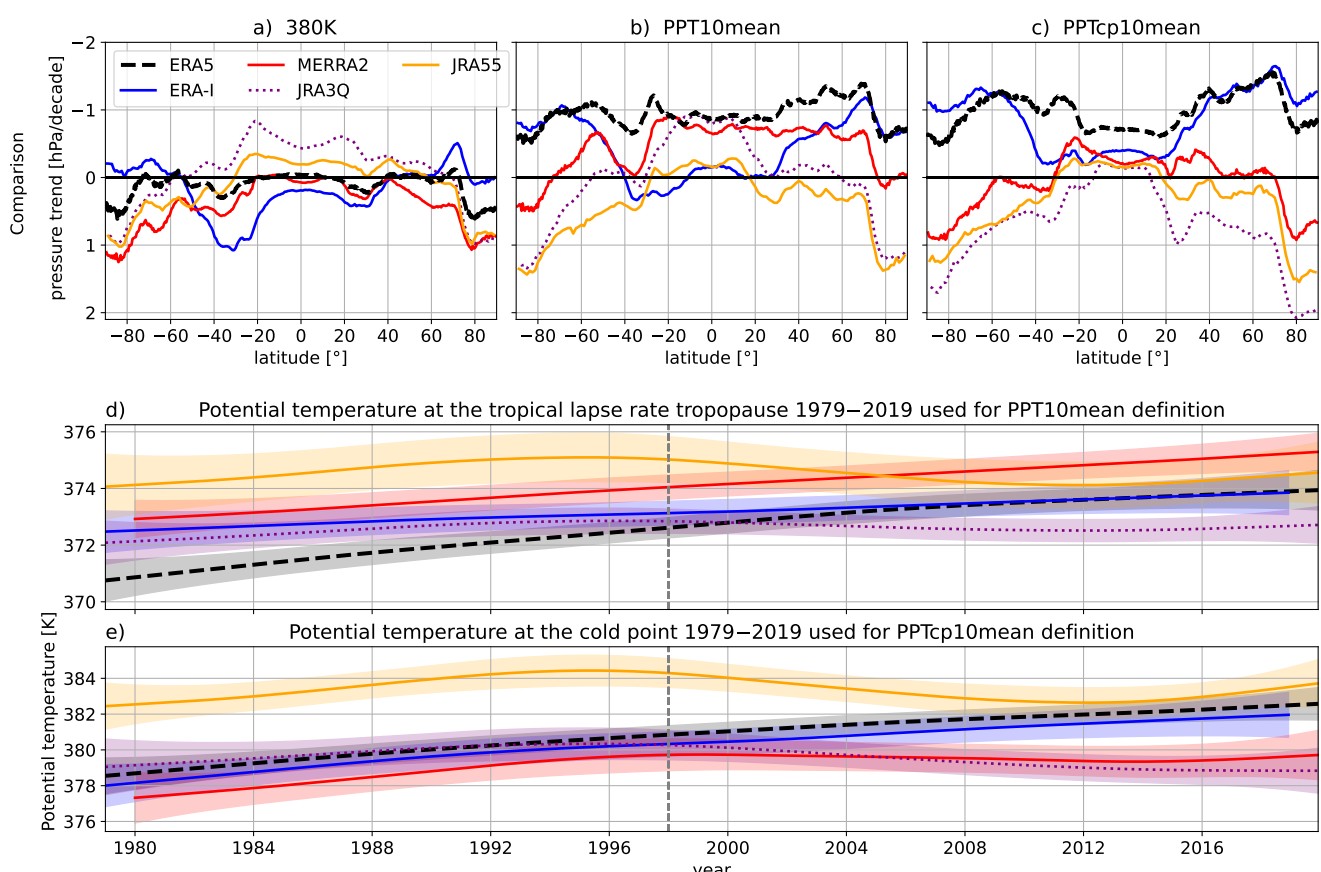

**Figure 10.** Similar to Fig. 9 but comparing the LMS upper boundary evolution across the reanalyses, i.e. ERA5 (black, dashed line), MERRA-2 (red, solid), ERA-Interim (blue, solid), JRA3Q (purple, dotted) and JRA-55 (orange, solid). In the upper panels, the mean DLM pressure trends (1998–2019) of the different LMS upper boundaries, i.e. 380 K (a), PPT10mean (b) and PPTcp10mean (c) are presented. For the sake of clarity, the presentation of uncertainty has been omitted in the upper panels but is of the same order of magnitude as in Fig. 9. Note that the pressure trend axis is inverted because decreasing pressure is associated with an upward trend in the pressure coordinate system. The lower panels show the DLM trend state time series of potential temperature at the tropical ($10°$N-$10°$S) lapse rate tropopause (PT10mean, d) and the cold point (PTcp10mean, e) (solid line) between 1979–2019. Depicted are the mean (lines) together with the range of two standard deviations (shading). The grey vertical line marks the beginning of the year 1998.

associated with the tropical tropopause. Specifically, in ERA5, the dynamically defined upper boundary surfaces (PPT10mean and PPTcp10mean) show an opposite trend to the $380$ K isentrope (Fig. 9a–c). This is due to the fact that the potential temper-

340   ature at the tropical tropopause exhibits a positive trend (Fig. 9d). For the mean potential temperature at the ERA5 lapse rate tropopause between $10°$N–$10°$S, the DLM estimates an increase of $1.33 \pm 0.40$ K between 1998–2019. The temporal evolution

of the potential temperature corresponding to the cold point amounts to $1.72\pm0.53\,\mathrm{K}$ for the same time period. The potential temperature increase at the tropical tropopause leads to a global lifting of the defined upper LMS boundary surfaces in ERA5 (Fig. 9b and c). The upward trends of the PPT10mean and PPTcp10mean surfaces in ERA5 are statistically significant for most latitudes in the tropics and mid-latitudes. The average pressure trends of around $-1\,\mathrm{hPa/decade}$ are very similar to the trend of the 2 PVU tropopause and the lapse rate tropopause in the NH mid-latitudes (Fig. 2a and b). In the SH extratropics, the sign of the lapse rate tropopause pressure trend is opposite to that of the PPT surfaces.

Like in ERA5, a significant rise of the PPT10mean surface in the extratropics is also evident in ERA-I and MERRA2 (Fig. 10b). This can be attributed the increase of tropical tropopause potential temperatures, which define the isentropes corresponding to PPT10mean (Fig. 10d). JRA-55 and JRA3Q on, the other hand, suggest the opposite (Fig. 10b and d). At the cold point, ERA5 and ERA-Interim agree on continuously increasing potential temperatures between 1998–2019, whereas the JRA data sets show mostly decreasing potential temperatures and MERRA-2 almost no trend at all (Fig. 10e). This is reflected in the trends of the PPTcp10mean surface (Fig. 10c). The contrasting behavior of the reanalyses can at least partly be attributed to the respective temperature trends in the TTL region. As evident from Fig. 4, ERA5 and MERRA-2 suggest no temperature trend at $100\,\mathrm{hPa}$ between 1998–2019, whereas temperatures in JRA-55 and JRA3Q are decreasing by $-0.4\pm0.2\,\mathrm{K}$ and $-0.7\pm0.2\,\mathrm{K}$, respectively. In ERA-Interim, temperatures at $100\,\mathrm{hPa}$ are significantly increasing after 1998. For the LMS mass analysis in Sect. 3.3.2 and 3.3.3, we use $380\,\mathrm{K}$, PPT10mean and PTcp10mean as upper LMS boundaries and compare their respective effects on the LMS mass. The different trends of the iso-surfaces discussed here are reflected in the LMS mass trends.

Like the upper and lower LMS boundaries, the boundaries separating the LMS laterally from the TTL-region are allowed to vary dynamically with time. For this purpose, the intersections between the monthly mean tropopause and the $350\,\mathrm{K}$ isentrope are identified via the sign change of the pressure difference between both surfaces. This intersection serves as a proxy for the position of the subtropical jet stream (e.g., Manney et al., 2011; Gettelman et al., 2011). Specifically, the lapse rate tropopause and the 2 PVU surface at $350\,\mathrm{K}$ are close to the isentropic PV-gradient tropopause (Fig. 8), which marks a clear transport barrier between the tropical troposphere and the extratropical lower stratosphere, i.e. the LMS (Kunz et al., 2011; Turhal et al., 2024). For comparison, the intersections of the tropopause with isentropes between 330–370 K are determined in addition. The $380\,\mathrm{K}$ isentrope often does not intersect with the respective tropopause, which is why it is not further considered. The zonal mean latitudes of the intersection between the lapse rate (2 PVU) tropopause and the $350\,\mathrm{K}$ isentrope, i.e. the lateral LMS boundaries in ERA5, vary between 29°N–48°N (26°N–40°N) and 31°S–45°S (26°S–36°S), with the lapse rate tropopause intersection always poleward of the 2 PVU intersection. Other definitions of the lateral LMS boundaries have been used in the literature like the intersection between the respective upper and lower LMS boundary surfaces (Appenzeller et al., 1996; Hegglin et al., 2010) or the zero diabatic heating rate at the upper LMS boundary surface (Wang and Fu, 2021; Wang et al., 2022). We consider the latitude of the intersection between tropopause and the $350\,\mathrm{K}$ isentrope to be a practical approximation of the location of the physical transport barrier between the (extratropical) LMS and the tropical troposphere, marked by the isentropic PV-gradient tropopause.

Beside the seasonal variability, the DLM identifies a trend of the intersections between tropopause and $350\,\mathrm{K}$ isentrope as

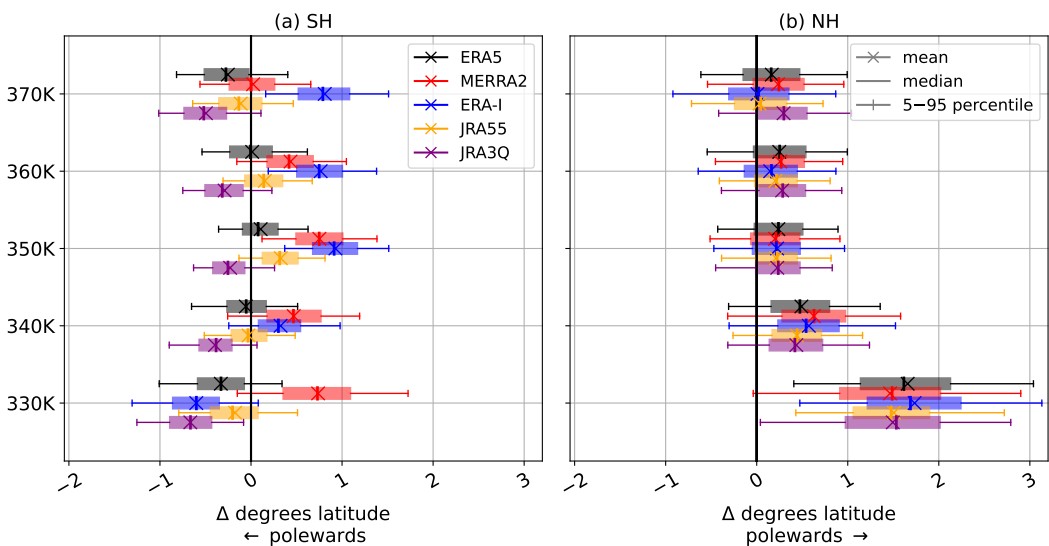

**Figure 11.** Trend of the latitude of intersection between the zonal mean lapse rate tropopause and isentropes between 330–370 K (y-axis) for all reanalyses. The reanalyses compared are ERA5 (black, uppermost box-plot), MERRA-2 (red, second box), ERA-Interim (blue, third box), JRA-55 (yellow, fourth box) and JRA3Q (purple, lowermost box-plot). The trends are DLM estimates for the time period 1998–2019. In the SH (a), negative trends are associated with a poleward tendency, in the NH (b), positive trends indicate a poleward shift. The DLM trend estimates are presented as box-whisker plots.

shown in Fig. 11. Here, the intersections of the zonal mean tropopause and isentropes are considered. In the NH, this trend is directed poleward for the lapse rate tropopause in all reanalyses (Fig. 11b) as well as 2 PVU in ERA5 (Fig. A5b). Such a poleward tendency is also evident for intersections of the NH tropopause with isentropes at 360 K and 370 K, which indi-
380   cates an expansion of the NH tropics. The results are robust across all examined reanalyses and consistent with evidence for a widening of the tropical belt, reported in many studies (e.g., Birner, 2010a; Wilcox et al., 2012; Xian and Homeyer, 2019). Our metric and findings are similar to other metrics based on cross-points between isentropic and potential vorticity fields (J. Añel, personal communication).

In the SH, on the other hand, the lateral tendencies of the intersections between tropopause and isentropes are less clear. For
385   the lapse rate tropopause at 350 K, all reanalyses except JRA3Q agree on an equatorward trend, however differing in magni-tude and statistical significance. In contrast to the lapse rate tropopause, the 2 PVU dynamical tropopause in ERA5 shows a poleward trend at 350 K (Fig. A5a). The general shape of the lateral troppause trends with respect to potential temperature, however, is similar in most reanalyses (Fig. 11a) and has also been found for the PV-gradient tropopause by Turhal et al. (2024). The different magnitudes of the lateral trends for tropopause intersections with different isentropes suggests a shape change
390   of the subtropical tropopause. The larger poleward tendency of the tropopause around 370 K, as evident in ERA5, JRA55 and

JRA3Q, hints at a steeper tropopause. This could be associated with higher zonal wind speeds, i.e. an intensification of the subtropical jet (Manney and Hegglin, 2018; Maher et al., 2020). Furthermore, such a steepening of the subtropical tropopause, including the tropopause break, could be an indication of a strengthened BDC (Birner, 2010b).

As far as the LMS mass is concerned, the temporal evolution of the lower, upper and lateral LMS boundaries points towards a trend of the LMS mass, which will be addressed in Sect. 3.3.3.

### 3.3.2 LMS mass climatology

The mean LMS mass in ERA5 between all considered boundary surfaces on both hemispheres is listed in Tab. 2. In ERA5, the average global LMS mass between the different boundary surfaces (lrtp, $2\,\mathrm{PVU}$ and $380\,\mathrm{K}$, PPT10mean, PPTcp10mean) ranges from $0.99\pm0.28\cdot10^{17}\,\mathrm{kg}$ (resulting from lrtp-PPT10mean) to $1.89\pm0.37\cdot10^{17}\,\mathrm{kg}$ (resulting from $2\,\mathrm{PVU}$-PPTcp10mean) within the time period 1979–2019. Since the $2\,\mathrm{PVU}$ tropopause is located at significantly higher pressure than the lapse rate tropopause (Fig. 8), the LMS mass between $2\,\mathrm{PVU}$ and the respective upper boundary is 1.6–1.8 times larger than the mass with respect to the thermal tropopause. The LMS mass with respect to the different upper boundary surfaces varies by less than $15\,\%$ for the same lower boundary and hemisphere. If the upper edge of the stratosphere is approximated at $1\,\mathrm{hPa}$, the proportion of LMS mass to total stratospheric mass is about $20\,\%$.

LMS mass values in ERA-Interim, MERRA-2, JRA-55 and JRA3Q are very similar and the different boundary surfaces have the same effect as in ERA5 (Tab. A1). The MERRA-2 LMS mass deviates the most from the other reanalyses, generally showing smaller values but the relative differences between the data sets are smaller than $10\,\%$ in the NH and smaller than $20\,\%$ in the SH. The variations of LMS mass across the reanalyses can be primarily attributed to the differences in tropopause location, determining the lower LMS boundary as well as the dynamical upper boundaries (Fig. A6).

In general, the NH mean LMS mass is smaller than the SH mass, independent of the boundary surfaces. This is due to the greater seasonal variability of the LMS mass in the NH, reaching smaller minima in summer than in the SH. Within one hemisphere, low latitudes contribute considerably more to the LMS mass than high latitudes due to the area decrease from equator to pole.

The LMS mass calculated in this study between the respective boundary surfaces can be compared to results reported by Appenzeller et al. (1996) and Hegglin et al. (2010) (Tab. 2). Using the example of ERA5, the LMS mass in this study is on average 1.5 to 4.8 times smaller than previously reported. Reason for the discrepancies are especially the choice of the boundaries. Appenzeller et al. (1996) integrate the pressure between the $2\,\mathrm{PVU}$ surface and the $380\,\mathrm{K}$ isentrope from UKMO (United Kingdom Meteorological Office) data with a horizontal grid of $2.5°$ latitude by $3.75°$ longitude and $50\,\mathrm{hPa}$ in the vertical, with zonal mean daily data for the year 1993. The resulting UKMO dynamic tropopause is situated around 50–100 hPa lower than the ERA5 $2\,\mathrm{PVU}$ surface. Hegglin et al. (2010) use NCEP (National Centers for Environmental Prediction) monthly mean, zonal mean reanalysis data for the time period 1990–1999 with a horizontal grid of $2.5°$ by $2.5°$. They choose the thermal tropopause as the lower and the $100\,\mathrm{hPa}$ isobar as a fixed upper boundary for the mass computation. The upper boundaries used in this study are (on average) situated between 90–187 hPa. For the lateral boundaries, Hegglin et al. (2010) and Appenzeller et al. (1996) use the intersections between upper and lower boundary surfaces. The lateral LMS boundaries in this study are at

| study and data | LMS | NH | SH |
|---|---|---|---|
| this study: | | | |
| ERA5 | lrtp-PPT10mean | 0.49±0.20 | 0.50±0.08 |
| | lrtp-PPTcp10mean | 0.56±0.22 | 0.58±0.09 |
| | lrtp-380K | 0.55±0.20 | 0.57±0.10 |
| | 2PVU-PPT10mean | 0.80±0.24 | 0.90±0.11 |
| | 2PVU-PPTcp10mean | 0.89±0.25 | 1.00±0.12 |
| | 2PVU-380K | 0.88±0.23 | 0.99±0.13 |
| Appenzeller et al. (1996): | | | |
| UKMO | 2 PVU–380 K | 2.3±0.4 | 2.4±0.1 |
| Hegglin et al. (2010): | | | |
| NCEP | lrtp–100 hPa | 1.6±0.3 | 1.5±0.1 |

**Table 2.** Mean LMS mass and associated standard deviation in $10^{17}$ kg for different LMS definitions and data. As an example for LMS mass in this study, LMS mass from ERA5 data is averaged over 1979–2019 and rounded to the second decimal. In Appenzeller et al. (1996) Fig. 5, daily LMS mass values from UKMO data are presented for the years 1992 and 1993. Fig. 6 in Hegglin et al. (2010) shows monthly mean LMS mass from NCEP data averaged over the time period 1990–1999. The LMS mass values from Appenzeller et al. (1996) and Hegglin et al. (2010) in this table are estimated from the respective figures

considerably higher latitudes. Using the same LMS boundary definitions as Hegglin et al. (2010) and ERA5 zonal mean data for the time period 1979–2019, we compute an average LMS mass of around $1 \cdot 10^{17}$ kg for each hemisphere. With a downward shift of the 2 PVU surface by 60 hPa, we obtain the same LMS mass as Appenzeller et al. (1996).

### 3.3.3 LMS mass trends

As presented in Sect. 3.1 and 3.3.1, the trends of the LMS lower, upper and lateral boundaries suggest a long-term change of the LMS mass enclosed by the respective boundary surfaces. In ERA5, the observed extratropical tropopause trends (Sect. 3.1, Fig. 2) and the lifting trend of the tropical tropopause (Fig. 9) indicate that the rise of the upper LMS boundary could potentially compensate for the rising lower boundary and the poleward transition of the lateral boundary. The results illustrated in Fig. 12a, b and d confirm this expectation. For the LMS in ERA5 bounded with 380 K, the DLM analysis results in a significant negative trend of the LMS mass between 1998–2019 (except for the LMS mass with respect to SH lapse rate tropopause). In contrast, the LMS defined with a dynamical upper boundary based on the tropical tropopause (PPT10mean and PPTcp10mean) shows mean mass trends close to zero or even positive trends. Focusing on the latitude bins poleward of the mean lateral LMS boundary (latitude bins 50°–80° and 60°–90°), LMS mass changes are positive or smaller negative but stay significantly negative if 380 K is used as the upper boundary. This suggests three things: 1) The decreasing NH LMS mass for a fixed 380 K boundary is largely due to the rising NH extratropical tropopause. 2) The "dynamical" upper boundary surfaces based on the tropical tropopause are indeed able to largely compensate for the tropopause rise in ERA5. 3) Locally, a

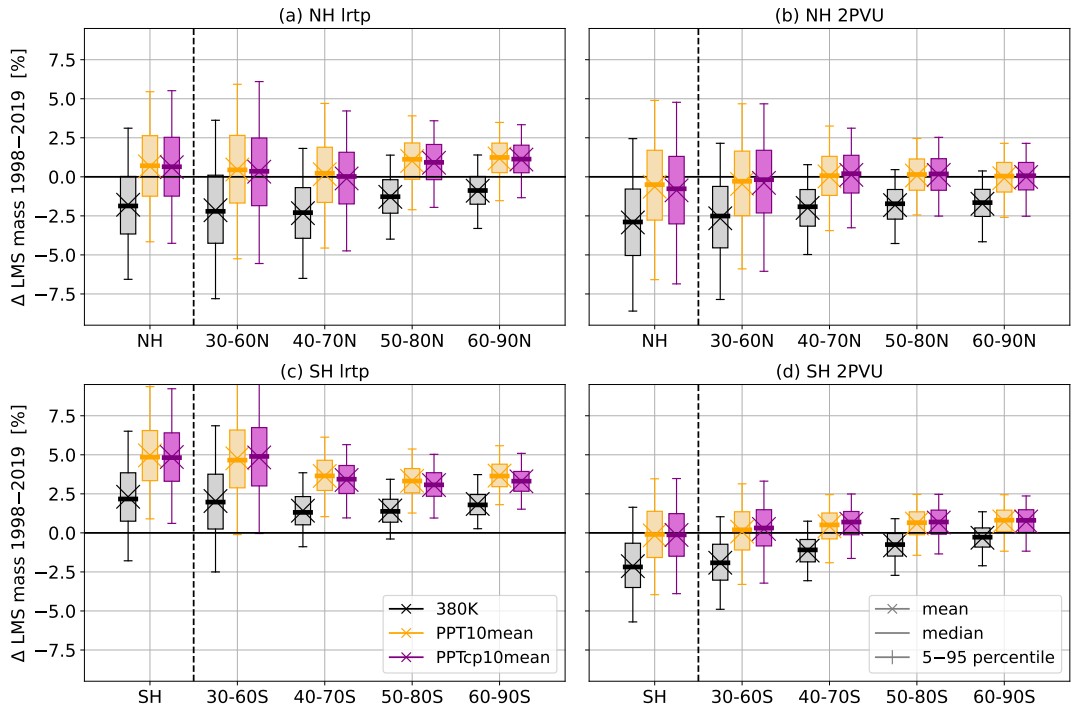

**Figure 12.** LMS mass change between 1998–2019 as estimated by the DLM from ERA5 data. The DLM trend state differences are presented as box-whisker plots. The LMS mass has been analyzed for different latitude bins (x-axis) for both hemispheres (upper and lower panels). The dashed vertical line separates the hemispheric trends from individual latitude bins. Moreover, different boundary surfaces have been compared. Panels (a) and (c) show the results for the LMS defined with the lapse rate tropopause as the lower boundary. In panels (b) and (d), the 2 PVU tropopause serves as the lower LMS edge. The results for the different upper boundary surfaces can be distinguished by the color of the box-whisker plots (380 K: black, PPT10mean: yellow, PPTcp10mean: purple).

considerable proportion of this NH LMS mass decline can be attributed to the poleward trend of the lateral boundary, which is the result of an expansion of the tropics, while the hemispheric effect is rather small. Fig. 13a confirms these observations and helps to quantify the contributions of the respective boundary surfaces to the hemispheric LMS mass changes. The figure compares LMS mass changes for an LMS where all boundary surfaces are evolving in time (as before, Fig. 12) with an LMS with only one boundary surface evolving in time ("floating"), while the other two are fixed (here, the first year of the time series is repeated).

ERA-Interim, MERRA-2, JRA-55 and JRA3Q agree remarkably well on the mass decrease for the LMS defined between the lapse rate tropopause and 380 K (Fig. 14a–f). In ERA-Interim and MERRA-2, the dynamical upper LMS boundaries defined


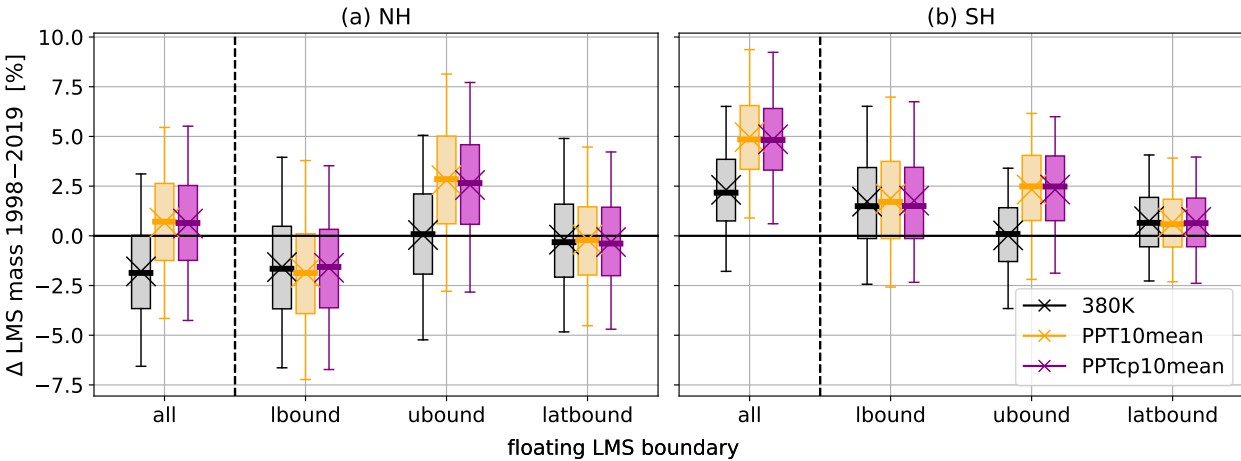

**Figure 13.** Similar to Fig. 12 but for hemispheric LMS mass difference if only one LMS boundary is allowed to evolve in time ("floating") while the other two are fixed, i.e. the year 1979 repeated. The respective boundaries are the lower boundary (lbound, here: the lapse rate tropopause), the upper boundary (ubound, either 380 K (black), PPT10mean (orange) or PTcp10mean (purple)) and the lateral boundary (latbound). For comparison, the LMS mass change between 1998–2019 for all boundary surfaces evolving in time is presented to the left of the dashed vertical line (same as in Fig. 12a and c).

with respect to the tropical tropopause have the same effect as in ERA5: They partly compensate for the tropopause rise. In JRA-55 and JRA3Q on the other hand, the dynamical boundary surfaces add to the LMS mass decline in the NH. This contrasting behavior can be attributed to the differing temperature trends at the tropical tropopause across the reanalyses. Such differing (potential) temperature trends are not only visible at the tropical lapse rate tropopause and the cold point (Figs. 3, 10d and e), but also at pressure surfaces like 100 hPa (Fig. 4). Thus, the differences in the LMS mass trends between the JRA data sets and the other reanalyses are rather rooted in the reanalysis data sets themselves than in the analysis methods used in this

study.

In Fig. 12c, the ERA5 LMS mass with respect to the SH lapse rate tropopause stands out with a significant LMS mass increase, regardless of the choice of the upper boundary. This is due to the increasing pressure trend of the SH extratropical lapse rate tropopause (Fig. 2a). Moreover, the intersection of the SH lapse rate tropopause with 350 K, which serves as the lateral

LMS boundary, shows a small equatorward tendency (Fig. 11a). Interesting are for example the respective LMS mass trends between 40°S–70°S (Fig. 12c). Since the ERA5 lapse rate tropopause change is close to zero between 40°S–70°S (Fig. 2a), the greater mass change for the SH LMS bounded with PPT10mean and PPTcp10mean compared to 380 K highlights the impact of the upper boundary surface, which is confirmed again in Fig. 13b.

In the SH, the mean LMS mass trends differ more strongly between the reanalyses than in the NH (Fig. 14g–l), especially for

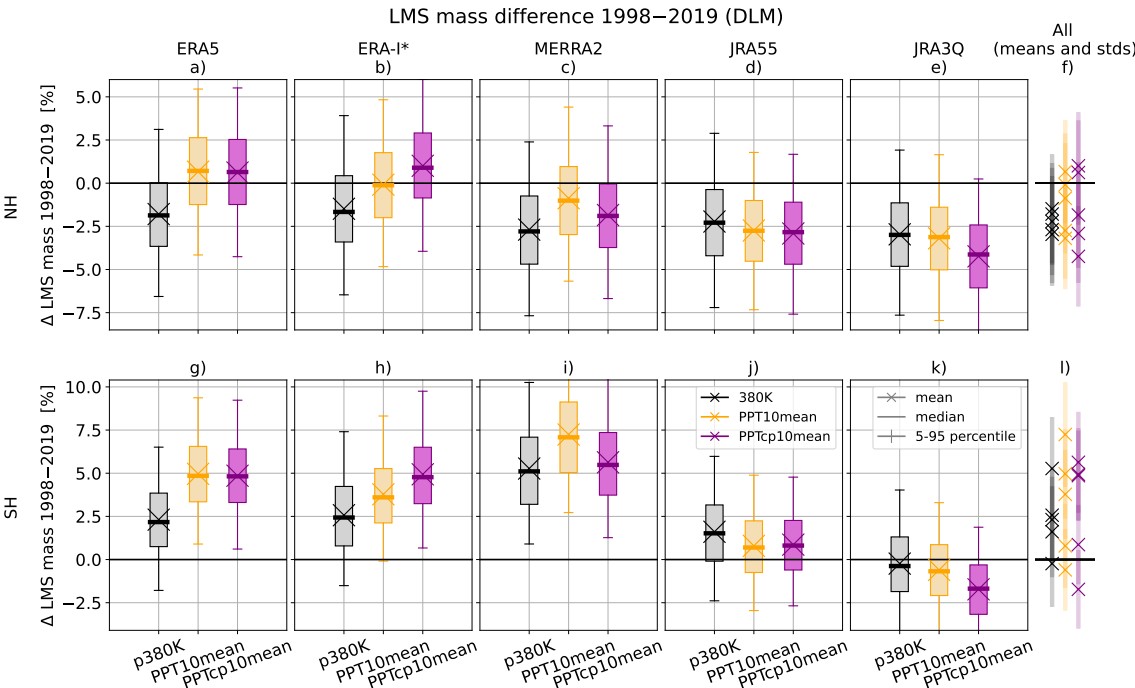

**Figure 14.** Similar to Fig. 12 but for LMS mass difference between 1998–2019 for different reanalyses (columns) on both hemispheres (rows). For better comparison, the mean trends (crosses) for all reanalyses are plotted again to the right of the figure (f and l), together with their associated standard deviation (transparent lines). The lapse rate tropopause defines the lower LMS boundary. *ERA-Interim time series ends in 2018

the LMS bounded by lrtp-380 K and lrtp-PPT10mean. Nevertheless, all reanalyses, except JRA3Q, suggest increasing LMS mass in the SH. In general, the LMS mass trends reflect the respective lapse rate tropopause trends suggested by the reanalyses (Fig. 3). Like in the NH, the use of the dynamical upper boundaries (PT10mean and PTcp10mean) has a positive effect on the LMS mass trends in ERA5, ERA-Interim and MERRA-2, whereas the opposite applies for JRA-55 and JRA3Q.

## 4    Conclusions

In this study, we have analyzed long-term trends in the thermodynamic structure of the lowermost stratosphere based on reanalysis data for the period 1979–2019. We compare results for five modern reanalyses: ERA5, ERA-Interim, MERRA-2, JRA-55 and JRA3Q.

Our DLM trend analysis shows that the zonal mean NH mid-latitude tropopause exhibits a negative pressure trend of about $-1\,\mathrm{hPa/decade}$, consistent with increasing height trends between 1998–2019. Although the tropopause trends lack strong sta-

tistically significance, they are very robust across the reanalysis data sets and in line with observations (e.g., Xian and Homeyer,

2019; Meng et al., 2021). In the tropics and subtropics, the ERA5 lapse rate tropopause and the cold point show a statistically significant upward trend between 1979–2019, evident in pressure and geopotential height. The pressure trend amounts to about $-0.5\,\mathrm{hPa/decade}$ at the tropical lapse rate tropopause and $-0.7\,\mathrm{hPa/decade}$ at the cold point. These tropopause trends agree well with trends reported in other studies of reanalysis data (e.g., Wilcox et al., 2012; Xian and Homeyer, 2019) and

observations (e.g., Meng et al., 2021). While all reanalyses examined in this study agree on decreasing tropopause pressure in the tropics, temperature trends in the TTL region are less consistent between the data sets. For ERA5, we find potential temperatures at the tropical tropopause to be significantly increasing, consistent with results from ERA-Interim and MERRA-2. JRA-55 and JRA3Q, on the other hand, suggest the opposite tendency.

The tendency of the SH tropopause in this study is less clear. In ERA5, the SH lapse rate tropopause shows a downward trend,

accompanied by pressure trends ranging between 0 and $+2\,\mathrm{hPa/decade}$, whereas the $2\,\mathrm{PVU}$ surface is rising at a mean rate of $-1\,\mathrm{hPa/decade}$. A positive pressure trend of the SH extratropical lapse rate tropopause is also evident in MERRA-2, JRA-55 as well as ERA-Interim and agrees with the trends found by Xian and Homeyer (2019) for the JRA-55, MERRA-2 and CFSR reanalyses. However, this positive pressure trend of the SH extratropical lapse rate tropopause seems to contradict the results reported for ERA-Interim in the same study and by Wilcox et al. (2012). Interestingly, our non-linear DLM trend analysis of

the time series 1979–2019 reveals a trend reversal of the lapse rate tropopause around the year 2000. Accordingly, our results suggest decreasing pressure of the SH mid-latitude lapse rate tropopause before the year 2000 and increasing pressure thereafter. This could be consistent with expected effects of ozone recovery and an accelerating BDC on tropopause height in the SH (Birner, 2010b). An acceleration of the BDC in the SH after about the year 2000 has also been estimated from age of air and long-lived trace gas measurements (Strahan et al., 2020; Ploeger and Garny, 2022).


The lateral positions of the intersections between the NH tropopause and isentropes between 330–370 K show a poleward shift after 1998, consistent across all reanalyses. The poleward tendency of the intersections can be associated with a widening of the tropics in the NH, which has been reported in many studies (e.g., Seidel and Randel, 2007; Meng et al., 2021; Wilcox et al., 2012; Grise et al., 2019; Staten et al., 2018). In the SH, the lateral trends of the intersections between lapse rate tropopause

and 350 K are directed equatorwards, while 330 K and 370 K show a poleward tendency, even though not all reanalyses agree in this case. The different magnitude of the lateral trends with respect to the potential temperature on both hemispheres suggests a steepening of the subtropical tropopause. A similar shape of lateral troopause trends with respect to potential temperature has been found for the PV-gradient tropopause by Turhal et al. (2024). Such a steepening of the subtropical tropopause, including the tropopause break, can be associated with an intensification of the SH subtropical jet (Manney and Hegglin, 2018; Maher

et al., 2020) and could be linked to a strengthening of the BDC (Birner, 2010b).

Since the NH tropopause is rising and expanding towards the poles, the mass of the lowermost stratosphere (LMS) can be expected to decrease if the upper boundary is fixed. This expectation is confirmed by the robust mass decline of the northern hemispheric LMS bounded by the $380\,\mathrm{K}$ isentrope, which amounts to $2$–$3\,\%$ between 1998–2019, according to this study.

However, as the upper LMS boundary is properly defined by the tropical tropopause, which is also rising, the negative LMS

mass trend is significantly reduced or even disappears when this dynamic upper boundary is used in the calculation. This is at least the case for ERA5, ERA-Interim and MERRA-2. JRA-55 and JRA3Q show opposite temperature trends at the tropical tropopause. This by itself is an important finding, since TTL temperatures are a critical variable, not only impacting LMS mass, but also confining lower stratospheric water vapor (e.g., Randel et al., 2004) and influencing stratospheric dynamics (e.g.,

Charlesworth et al., 2023). Why the reanalyses show the aforementioned discrepancies is beyond the scope of this study but should be investigated further. Hints can be found in Fujiwara et al. (2022) and Fujiwara et al. (2024), suggesting relationships between differences in tropical lower stratospheric temperature and radiative heating, related to ozone concentrations, among other factors.

Due to the downward trend of the SH lapse rate tropopause, evident in all reanalyses except JRA3Q, which is accompanied by

increasing pressure, the LMS mass in the SH is found to have increased by 1–7 % between 1998–2019. The different upper boundary surfaces have the same effect on the LMS mass in the SH as in the NH: adding to the mass increase in ERA5, ERA-Interim and MERRA-2 but reducing it in JRA-55 and JRA3Q.

In general, the vertical boundaries dominate LMS mass changes. However, tropical width changes can have a considerable impact on LMS mass due to the area decrease from low to high latitudes.


LMS mass changes are important to consider because the LMS mass makes up a considerable amount of the stratospheric column, in our case approximately 20 %, an therefore contains a substantial fraction of, for example, column ozone. Since the interannual ozone variability in the LMS is mainly controlled by dynamics (e.g., Chipperfield et al., 2018), ozone changes are expected to be directly related to LMS mass changes. LMS ozone changes do not only have important implications on the

radiative budget in the lower stratosphere but also on atmospheric chemistry and the UV radiation reaching the surface (e.g., Forster and Shine, 1997; Hegglin and Shepherd, 2009).

*Code and data availability.* ERA5 and ERA-Interim reanalysis data are available from the European Centre for Medium-range Weather Forecasts (ECMWF (b), 2024) and (ECMWF (a), 2024). The MERRA-2 data set is provided by Global Modeling And Assimilation Office and Pawson (2015) of the National Aeronautics and Space Administration (NASA). The JRA-55 reanalysis data is available at Japan Mete-

orological Agency/Japan (2013) and the JRA3Q data set at Japan Meteorological Agency/Japan (2024). The DLM model code (dlmmc) is publicly available at https://github.com/justinalsing/dlmmc. The code for LMS mass calculation and trend analysis, including the respective LMS boundary surfaces in all reanalyses, is available on zenodo (Weyland, 2024).

*Author contributions.* FW performed the analyses and wrote the first draft of the paper. PH and DK initiated the project and conceptualized the core research goals. FP and TB contributed valuable ideas. KT provided the PV-gradient tropopause data. All authors contributed in

interpreting the results and improving the manuscript.

*Competing interests.* The authors declare that they have no conflict of interest.

*Acknowledgements.* This work was funded by the Deutsche Forschungsgemeinschaft (DFG, German Research Foundation) – TRR 301 – Project-ID 428312742: "The tropopause region in a changing atmosphere". We further would like to thank the Dres. Göbel Klima-Stiftung for their financial support, which FW was able to benefit from as part of a Deutschlandstipendium during her Master's programme.

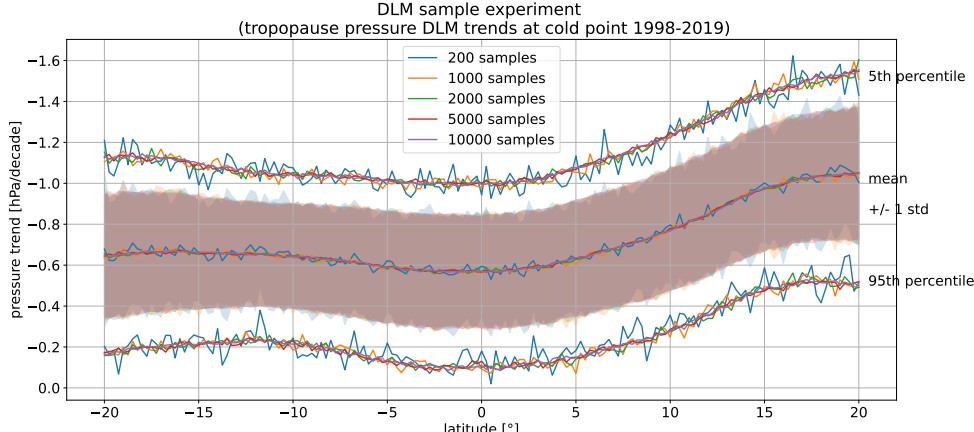

**Figure A1.** Illustration of the sensitivity of DLM trend estimates on the number of DLM samples using the example of cold point pressure trends between 1998–2019.

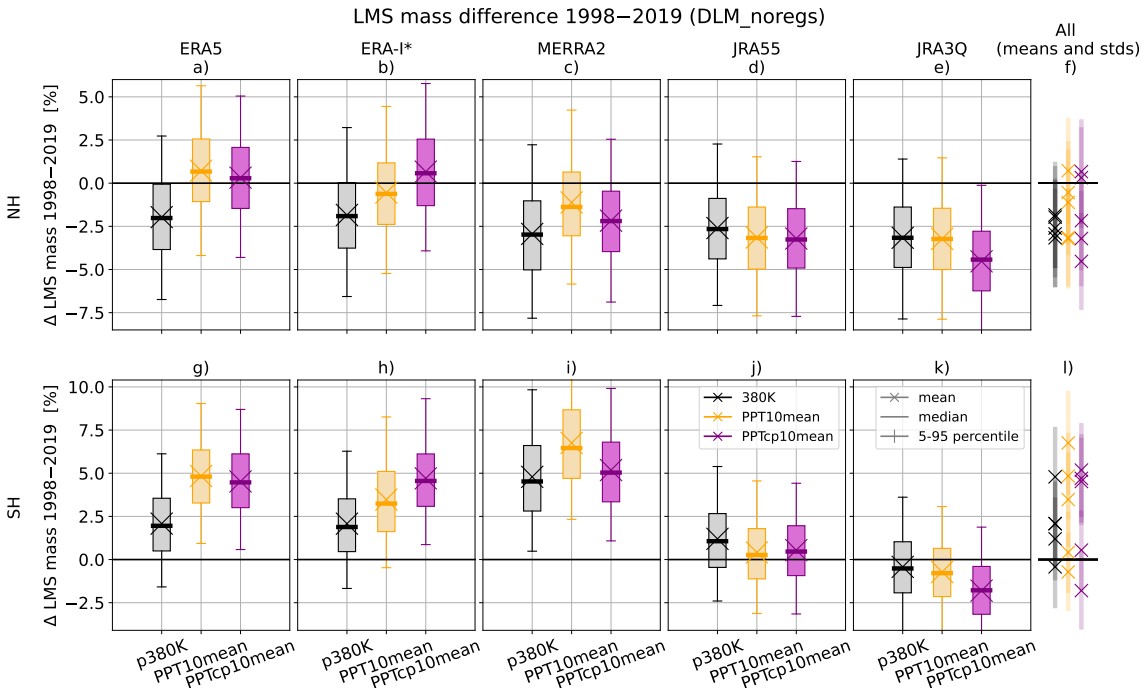

**Figure A2.** Same as Fig. 14 but for DLM analysis without regressors.

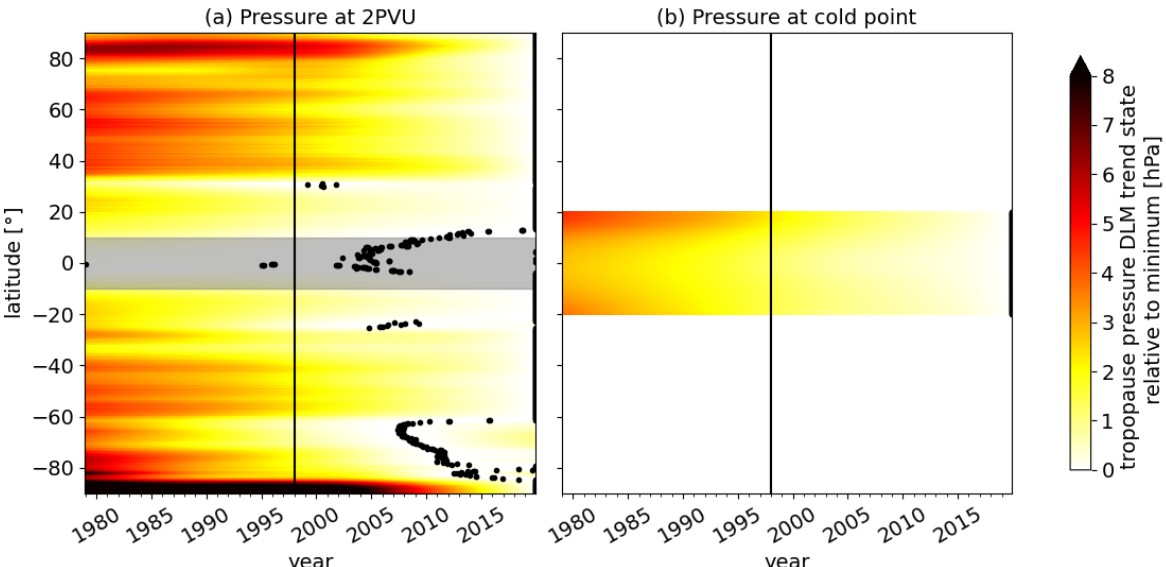

**Figure A3.** Same as Fig. 5 but for the ERA5 2 PVU dynamic tropopause (a) and the cold point (b). Note that since the 2 PVU iso-surface is capped at a fixed pressure of 89 hPa in the inner tropics, the respective pressure trends are not meaningful (grey shading).

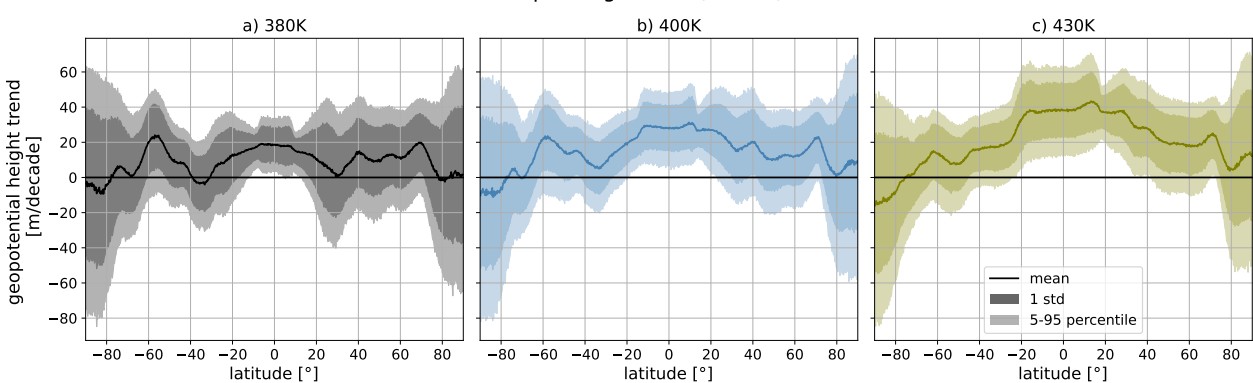

**Figure A4.** Same as Fig. 6 a–c but for geopotential height trends.

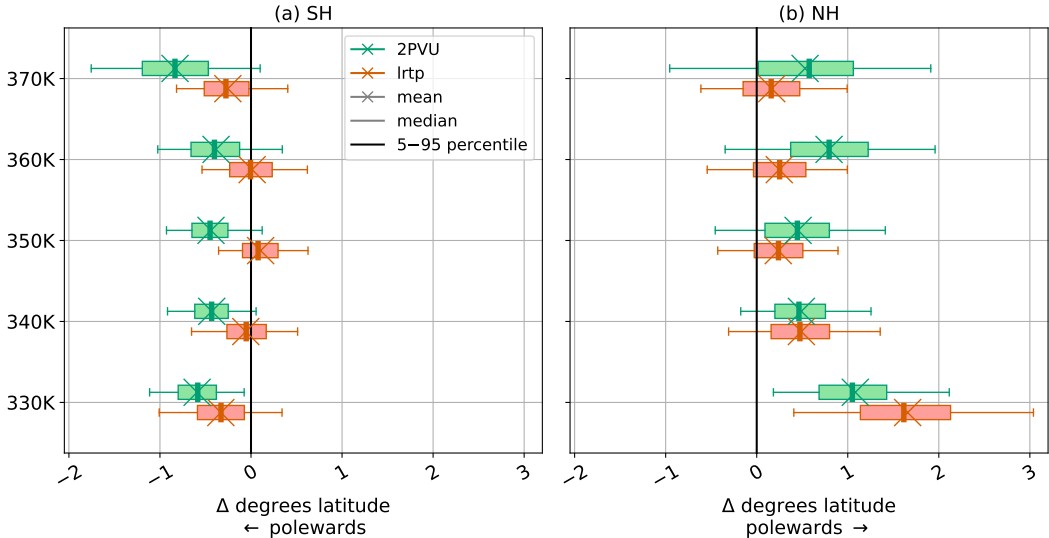

**Figure A5.** Same as Fig. 11 but for the intersections between isentropes with the lapse rate tropopause (red) and the $2\,\mathrm{PVU}$ dynamic tropopause in ERA5.

| LMS definition | Reanalysis | NH | SH |
|---|---|---|---|
| lrtp-380K | ERA5 | 0.55±0.20 | 0.57±0.10 |
| | MERRA-2 | 0.50±0.20 | 0.51±0.11 |
| | ERA-Interim | 0.54±0.20 | 0.57±0.11 |
| | JRA-55 | 0.53±0.19 | 0.56±0.10 |
| | JRA3Q | 0.54±0.19 | 0.57±0.10 |
| lrtp-PPT10mean | ERA5 | 0.49±0.20 | 0.50±0.08 |
| | MERRA-2 | 0.46±0.19 | 0.46±0.10 |
| | ERA-Interim | 0.49±0.20 | 0.51±0.09 |
| | JRA-55 | 0.49±0.19 | 0.51±0.09 |
| | JRA3Q | 0.49±0.18 | 0.51±0.08 |
| lrtp-PPTcp10mean | ERA5 | 0.56±0.22 | 0.58±0.09 |
| | MERRA-2 | 0.49±0.21 | 0.50±0.10 |
| | ERA-Interim | 0.55±0.22 | 0.57±0.10 |
| | JRA-55 | 0.56±0.21 | 0.58±0.09 |
| | JRA3Q | 0.54±0.20 | 0.57±0.08 |

**Table A1.** Mean LMS mass and associated standard deviation in $10^{17}$ kg for different LMS definitions and reanalysis data sets.

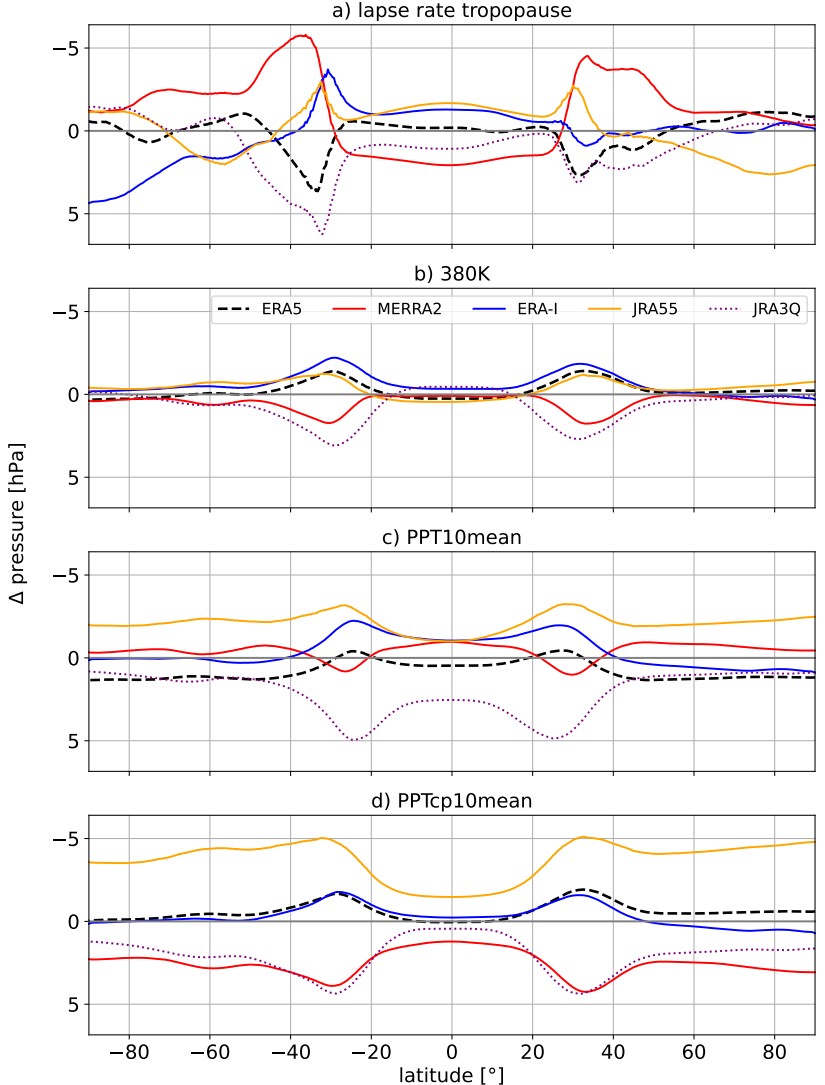

**Figure A6.** Illutration of the differences in the location of the LMS boundary surfaces, i.e. the lapse rate tropopause (a), 380 K (b), PPT10mean (c) and PPTcp10mean (d) across the reanalyses (ERA5, black, dashed line; MERRA-2, red, solid; ERA-Interim, blue, solid; JRA-55, orange, solid; JRA3Q, purple, dotted). Shown are the deviations from the multi-reanalysis mean in hPa. Note that the y-axis is inverted because pressure is decreasing with altitude. ERA5, JRA-55 and JRA3Q span the time period 1979–2019, MERRA-2 the time period 1980–2019 and ERA-Interim the time period 1979–2018.

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
