# Peer review of "Long-term changes in the thermodynamic structure of the lowermost stratosphere inferred from reanalysis data"

_EGUsphere, 2024_

## Referee Comment (RC2)

**Tropical expansion measured by isentropic and potential vorticity fields and age of air**

**Juan A. Añel[1], Petr Šácha[1] and Laura de la Torre[1]**

j.anhel@uvigo.es

[Figure]

UniversidadeVigo

EPhysLab

[1]**Ourense, Spain**

**Some methods used to evaluate tropical width**

- SSTs
- subtropical jet stream/barriers
- tropopause height - with 'artificial' thresholds -
- surface winds
- outgoing IR radiation - maximum at subtropics
- precipitation-evaporation difference
- $[O_3]$ change tropics-subtropics

[Figure]

Birner et al.(2014) adapted from IPCC AR5

**Problems assessing tropical width**

- Some methodologies can only be applied to a given subset
- Some variables are very poor from models limiting future projections
- Most of them depend on randomly chosen threshold values
- Tropical widening is a problem more complex that getting a single 2D perspective

**Global Distribution of Potential Vorticity at the tropopause level**

[Figure]

FIG. 1. Potential vorticity on the 320-K isentropic surface at 0000 GMT 15 February 1994, in lat–long projection for the Northern Hemisphere. Isolines are plotted every 1 PVU. Field produced by Dr. P. van Velthoven, using analyzed data from the European Centre for Medium-Range Weather Forecasts.

[Figure]

FIG. 2. Potential vorticity (in PVU) for the case of Fig. 1 plotted as a function of the area that is covered by potential vorticity values that are larger than the value at hand. Area is expressed as equivalent latitude, i.e., the latitude of a zonally symmetric contour enclosing the same area.

Ambaum M. (1997), Isentropic Formation of the Tropopause, J. Atmos. Sci., 54, 555-568.

[Figure]

**Data & Methodology**

- Analysis of the pressure leveles between 500 and 50 hPa.
- Datasets used: WACCM, WACCM UTLS, ERA-INTERIM, JRA55, NCEP2, MERRA

|  | N levels 500-50 hPa | Horizontal grid |
|---|---|---|
| **WACCM** | 14 | 1.9×2.5 |
| **WACCM UTLS** | 40 | 1.9×2.5 |
| **ERA-INTERIM** | 14 | 0.5×0.5 |
| **JRA55** | 14 | 1.25×1.25 |
| **MERRA** | 11 | 1.25×1.25 |
| **NCEP2** | 9 | 2.5×2.5 |

**Data & Methodology**

- $\theta$ is computed for each grid point

$$\theta(\lambda, \phi, p, t) = T(\lambda, \phi, p, t) \cdot (p_0/p)^{(R/c_p)}$$

- $u$, $v$ and $z$ fields are interpoled to the $\theta$ levels interesting for our study (from 340K to 420K)
- the isentropic relative vorticity of the air $\xi_\theta$ is computed
- then we have the absolute vorticity
- $d\theta/dp$ is computed and then interpoled to isentropic levels.
- PV is computed

$$PV(\lambda, \phi, \theta, t) = -g \cdot (\xi_\theta + f) \cdot (d\theta/dp)$$

- the PV field is interpoled to the desired values (1.6, 2, 3, 3.5, 4, 5, 6 PVU and the subsequent for the Southern Hemisphere).
- $\phi_e$ is computed

**Equivalent latitude**

[Figure]

Añel et al. (2013) Equivalent Latitude Computation Using Regions of Interest (ROI).
PLoS ONE 8(9): e72970

Period of study:

WACCM — 1979-2006
WACCM-UTLS — 1979-2006
ERA-INTERIM — 1979-2013
JRA55 — 1979-2013
MERRA — 1979-2013
NCEP2 — 1979-2013

$\phi_e$ **trend (degrees/decade) averaged for 340 K, 350 K and 360 K. Positive trends: northward displacement; negative trends: southward displacements**

|  | WACCM | WACCM-UTLS | ERA-Interim | MERRA | NCEP2 | JRA-55 |
|---|---|---|---|---|---|---|
| **-6.0** | $-0,27 \pm 0,11$ | **-0.04**±0,17 | **-0.17**±0,14 | **0.02**±0,13 | **0.72**±0,12 | **0.06**±0,16 |
| **-5.0** | $-0,27 \pm 0,09$ | **0.00**±0,17 | **0.08**±0,11 | **0.14**±0,10 | **0.66**±0,10 | **0.15**±0,11 |
| **-4.0** | $-0,24 \pm 0,08$ | **0.05**±0,18 | **-0.04**±0,11 | **-0.21**±0,07 | **-0.04**±0,05 | **-0.05**±0,06 |
| **-3.5** | $-0,23 \pm 0,08$ | **0.06**±0,18 | **0.19**±0,16 | **0.09**±0,08 | **0.87**±0,11 | **0.15**±0,11 |
| **-3.0** | $-0,21 \pm 0,09$ | **0.05**±0,17 | **-0.04**±0,08 | **0.00**±0,04 | **-0.84**±0,10 | **-0.19**±0,08 |
| **-2.0** | $-0,24 \pm 0,10$ | **-0.01**±0,18 | **0.00**±0,08 | **0.06**±0,05 | **-0.41**±0,08 | **-0.05**±0,09 |
| **-1.6** | $-0,29 \pm 0,12$ | **-0.08**±0,21 | **0.04**±0,10 | **0.16**±0,07 | **-0.16**±0,08 | **0.05**±0,09 |
| **1.6** | $0,38 \pm 0,13$ | **0.25**±0,17 | **0.41**±0,12 | **0.45**±0,12 | **-0.01**±0,09 | **0.23**±0,09 |
| **2.0** | $0,31 \pm 0,11$ | **0.23**±0,15 | **-0.27**±0,15 | **0.05**±0,14 | **0.76**±0,12 | **0.00**±0,14 |
| **3.0** | $0,19 \pm 0,09$ | **0.13**±0,11 | **0.13**±0,09 | **0.02**±0,09 | **0.47**±0,10 | **0.10**±0,09 |
| **3.5** | $0,15 \pm 0,08$ | **0.10**±0,10 | **-0.43**±0,15 | **-0.27**±0,09 | **0.10**±0,07 | **-0.20**±0,10 |
| **4.0** | $0,12 \pm 0,08$ | **0.06**±0,09 | **-0.04**±0,13 | **0.01**±0,08 | **0.59**±0,07 | **-0.02**±0,07 |
| **5.0** | $0,17 \pm 0,09$ | **0.03**±0,10 | **-0.10**±0,06 | **0.03**±0,04 | **-0.60**±0,08 | **-0.17**±0,06 |
| **6.0** | $0,23 \pm 0,12$ | **0.05**±0,12 | **0.04**±0,08 | **0.17**±0,05 | **-0.19**±0,07 | **0.02**±0,07 |

[Figure]

**WACCM REFB2**

[Figure]

[Figure]

**Number of days with AoA < mean value for 1979-2009. Source: CMAM**

[Figure]

**Period: 1979-2009. Source: CMAM**

[Figure]

**Conclusions**

- WACCM gives a clear poleward signal of the movement of the PV field
- Reanalysis give contradictory signals in several cases, but mostly agree for the isentropic levels closer to the tropopause
- The competing phenomena of tropopause rising vs broadening is clear.
- At UTLS levels the latitudinal gradient of AoA seems clear and average days under the mean could be an useful fingerprint

---

## Author Comment (AC1)

**Reply to Reviewer 1**

We thank the Reviewer for the careful reading and the constructive criticism of our manuscript. The comments and remarks have helped to improve the paper. We understand two main concerns raised by the Reviewer, regarding (i) the focus on one single reanalysis and (ii) equating pressure and height trends. We have extended our analysis and revised the manuscript accordingly. In the following, we address all comments in further detail (Reviewer's comments in italics, quotations of the corresponding revised text passages in blue).

Based on the suggestions of all reviewers, we have repeated our analyses of LMS structural changes with four modern reanalyses, namely ERA-Interim, MERRA-2, JRA-55 and JRA3Q, in addition to ERA5. The structure of the paper has essentially remained the same, although the individual sections have been expanded to include a comparison of the results between the reanalyses. We have also adapted the title accordingly: "Long-term changes in the thermodynamic structure of the lowermost stratosphere inferred from reanalysis data"

**Major comments**

**(i) Comparison of reanalysis data**

*I find the authors' decision to focus only on ERA5 without including other reanalyses in any way unfortunate. Although modern reanalyses are improving at a substantial rate, the differences between them, especially in the stratosphere, are well known. For example, for BDC, which the authors themselves mention as a very important factor for trends in LMS, a substantial spread in climatology and trends between reanalyses was found (e.g., Abalos et al. 2015 or Šácha et al., 2024). The inclusion of other reanalyses could make the conclusions substantially stronger in the case of agreement between reanalyses, or reveal discrepancies between them. In this case, where only one reanalysis is used, I recommend a substantial expansion of the discussion regarding the known differences between the most commonly used reanalyses.*

We agree that the significance of our study increases when comparing results across different reanalyses, as we are aware of the general strengths and limitations of reanalysis data and the differences between reanalysis data sets. We have therefore repeated all analyses with three widely used reanalyses ERA-Interim, MERRA-2, JRA-55 as well as the recently published JRA3Q data set. For the introduction and illustration of our metrics to assess the structure of the LMS, we use ERA5 before we generalize our findings to all other data sets.

**(ii) Pressure and height trends**

*Vertical shifts of tropopause and potential temperature levels in the paper are evaluated in the pressure coordinate system and the authors report trends in the pressures of these levels. However, the authors compare the detected trends for tropopause with previous studies from Xian and Homeyer (2019), Wilcox et al. (2012a) and Meng et al. (2021) in which trends are evaluated in a geometric vertical coordinate. Also in other parts of the paper (e.g., L215-216), the trends in pressure and altitude are interpreted as equivalent, which I do not consider to be correct as pressure levels in the troposphere shift upwards as a direct thermodynamic consequence of tropospheric warming (see, e.g., Fig. 1 from Eichinger and Šácha, 2020). So I'm concerned that comparing trends in geometric and pressure levels can be quite misleading. For example, if the pressure levels and the tropopause are moving upwards at the same rate, the tropopause pressure would not change, but the geometric height would. I would suggest redoing Figs. 2 and 4 in geometric coordinate, or the authors should show that trends in pressure level shifts are negligible compared to trends in tropopause/isentropic level height.*

We acknowledge that a direct comparison between (tropopause or isentropic) pressure and height trends can be misleading due to the mentioned thermodynamic effects. We have therefore made a clearer distinction between pressure and height trends in the manuscript. As suggested, we have added tropopause (geopotential) height trends to Fig 2. Nevertheless, our analysis of pressure and potential temperature at the tropopause and lower stratospheric isentropes was conducted to lead towards an understanding of the LMS mass trends presented. Potential temperature trends are relevant in this context because we define the

upper LMS boundary with respect to the potential temperature at the tropical tropopause. The pressure difference between upper and lower LMS boundary determines the LMS mass, whereas the (geometric or geopotential) altitude of theses surfaces plays no direct role in this context. Therefore, we do not discuss differences between pressure and height trends in further detail.

**Minor comments**

*L37: UTLS is not defined*

Thank you for the hint. The definition has been added.

*L94-95: papers Oberländer-Hayn et al., 2016 or Šácha et al, 2024 showed that tropical upwelling does not intensify at the tropopause, but at pressure levels.*

Thank you for the recommend reading. We understand that, according to Oberländer-Hayn et al. (2016) and Šácha et al. (2024), the expected acceleration of the BDC, including increasing tropical upwelling, is in fact to a large extent caused by a lifting of the circulation together with the tropopause (as well as other effects) instead of a pure acceleration of the vertical advection. This is an important information about BDC changes which has been added to the manuscript. Šácha et al. (2024) do report increased upwelling at the tropopause as well as the $100\,\mathrm{hPa}$ and $70\,\mathrm{hPa}$ in all reanalyses except ERA5, as evident from Fig. 3 in their study.

Line 96-98: According to Oberländer-Hayn et al. (2016) and Šácha et al. (2024), it is more precise to speak about a lifting of the circulation, which is connected to the tropopause expansion itself. Stratospheric temperature changes linked to stratospheric ozone additionally influence the BDC evolution.

*L109: Do you use ERA5.1 for the 2000-2006?*

Thank you for the reference to ERA5.1. The analysis had in fact been performed with ERA5 data so far. We have now updated all ERA5 analyses using ERA5.1 data for the time period 2000-2006. This update has only led to minor alterations of the trend results.

*L116: I suggest to change subsection 2.2 name to "Tropopause detection"*

„Tropopause detection" seems indeed to be a more suitable tittle for subsection 2.2. Changed accordingly.

*L117: cite ERA5 data*

Thank you for the hint. The sentence is now formulated more generally, due to the inclusion of additional reanalyses.

L155: To determine the lapse rate tropopause according to WMO (1957) from all five reanalysis data sets, [...]

*L117-215: The description of detection lapse rate tropopause is, in my opinion, insufficient. Although the authors refer to the meteorology (methodology) used in Birner, 2010, the clarity of the paper would benefit from a more detailed description of the meteorology (methodology) and data used.*

We agree that the (lapse rate) tropopause detection is a critical factor for UTLS studies. Despite its clear definition, identifying the lapse rate tropopause offers scope for different approaches in practice. We have now provided a more detailed description of our lapse rate detection algorithm, which we hope makes our methodology clearer.

L155-165: To determine the lapse rate tropopause according to WMO (1957) from all five reanalysis data sets, we apply an algorithm closely following that of Birner (2010a), based on the work of Reichler et al. (2003). The algorithm computes the temperature lapse rate on $p^{\kappa} = p^{R_d/c_p}$ half levels (where $p$ is

the pressure, $R_d$ is the specific gas constant and $c_p$ is the heat capacity at constant pressure for dry air), according to equations 1)–4) in Reichler et al. (2003). Subsequently, the algorithm identifies the lowest half level at which the lapse rate becomes smaller than $2\,\mathrm{K/km}$. Following Birner (2010a), a preliminary tropopause is then identified by linear interpolation of $p^\kappa$ and all other variables to a lapse rate of exactly $2\,\mathrm{K/km}$ (stratification threshold). Second, the algorithm checks whether the lapse rate remains on average below $2\,\mathrm{K/km}$ for all higher levels within $2\,\mathrm{km}$ (thickness criterion). Therefore, a temporal level at $2\,\mathrm{km}$ distance to the preliminary tropopause is added and the algorithm successively checks the average lapse rate for all levels between the preliminary tropopause and $2\,\mathrm{km}$ distance. If the preliminary tropopause does not fulfill the thickness criterion, the next higher half level fulfilling the stratification threshold is tested. [...]

*L125-126: I find this sentence very vague: The detected lapse rate tropopause agrees well with the ERAInterim lapse rate tropopause in Gettelman et al. (2010) and Wilcox et al. (2012b). If a quantitative comparison was made, why is there no metric given? Moreover, the comparison refers to ERAInterim, a reanalysis from the same family as ERA5. Has validation been performed using other reanalyses? Maybe authors could cite previous studies on this matter: e.g., Hoffmann and Spang, 2022 or Zou et al., 2023.*

The cited sentence was not ment to compare ERA5 and ERA-Interim lapse rate tropopauses in this case. The idea was to state that our lapse rate tropopause detection in ERA5 yields meaningful results. We admit that the sentence can be misunderstood and that this qualitative comparison is very vague. We have thus removed the sentence.

*L164: Has there been a strong dependence on the mentioned regressors? There is no discussion of the significance of the regressors throughout the paper.*

We conducted sensitivity test, running the DLM with and without regressors in order to check the robustness of the trends and their dependence on the regressors. However, our study does not aim to disentangle the contribution of different phenomena, represented by regression proxies, to the presented trends. Rather, we use the regressors for the sake of completeness and consistency with other studies on trends of UTLS characteristics (e.g. Seidel and Randel, 2006; Tegtmeier et al., 2020; Meng et al., 2021; Zou et al., 2023). We agree that a statement about the effect of the regressors was missing in the original manuscript. A short statement has now been added.

L215-220: [...] The same regressors have been used in different studies investigating changes of UTLS characteristics (e.g., Seidel and Randel, 2006; Meng et al., 2021; Tegtmeier et al., 2020; Zou et al., 2023). [...] In order to assess the robustness of the trends and the effect of the regressors, we conducted sensitivity tests in which the DLM was run with and without regressors. Overall, the results showed no strong dependency on the used regressors. However, cold point pressure trends, for example, become more significant when regressors are used (not shown).

*L166: SOAD is not defined*

Thank you for the hint. There has been a typo, it should be "SAOD". SAOD definition has been added.

*L184: potential temperature trend should be 0.7 K/decade*

Thank you! Changed.

*L325: long-term*

Done.

*L337-338: the effect of density may be more complex due to changes in both temperature and pressure in the LMS region.*

We recognize that we are not able to make a precise statement about density effects at this point. The point we want to make is: The contribution of atmospheric levels to the mass of an atmospheric layer

decreases with altitude, assuming an exponential density decrease with altitude. We have changed the sentence accordingly.

L424-425: The "dynamical" upper boundary surfaces based on the tropical tropopause are indeed able to largely compensate for the tropopause rise in ERA5.

**References**

Meng, L., Liu, J., Tarasick, D. W., Randel, W. J., Steiner, A. K., Wilhelmsen, H., Wang, L., and Haimberger, L.: Continuous rise of the tropopause in the Northern Hemisphere over 1980–2020, Science Advances, 7, eabi8065, https://doi.org/10.1126/sciadv.abi8065, 2021.

Oberländer-Hayn, S., Gerber, E. P., Abalichin, J., Akiyoshi, H., Kerschbaumer, A., Kubin, A., Kunze, M., Langematz, U., Meul, S., Michou, M., Morgenstern, O., and Oman, L. D.: Is the Brewer-Dobson circulation increasing or moving upward?, Geophysical Research Letters, 43, 1772–1779, https://doi.org/10.1002/2015GL067545, 2016.

Seidel, D. J. and Randel, W. J.: Variability and trends in the global tropopause estimated from radiosonde data, Journal of Geophysical Research, 111, D21 101, https://doi.org/10.1029/2006JD007363, 2006.

Tegtmeier, S., Anstey, J., Davis, S., Dragani, R., Harada, Y., Ivanciu, I., Pilch Kedzierski, R., Krüger, K., Legras, B., Long, C., Wang, J. S., Wargan, K., and Wright, J. S.: Temperature and tropopause characteristics from reanalyses data in the tropical tropopause layer, Atmospheric Chemistry and Physics, 20, 753–770, https://doi.org/10.5194/acp-20-753-2020, 2020.

Zou, L., Hoffmann, L., Müller, R., and Spang, R.: Variability and trends of the tropical tropopause derived from a 1980–2021 multi-reanalysis assessment, Frontiers in Earth Science, 11, 1177 502, https://doi.org/10.3389/feart.2023.1177502, 2023.

Šácha, P., Zajíček, R., Kuchař, A., Eichinger, R., Pišoft, P., and Rieder, H. E.: Disentangling the Advective Brewer-Dobson Circulation Change, Geophysical Research Letters, 51, e2023GL105 919, https://doi.org/10.1029/2023GL105919, 2024.

---

## Author Comment (AC2)

**Reply to Reviewer 2**

We appreciate the reviewer's interest in our study. However, we would like to clarify some apparent misunderstandings: The LMS mass is calculated based on the formalism of Appenzeller et al. (1996). However, our study does not address mass transport between the tropics and subtropics, nor does it utilize Transformed Eulerian Mean equations. Our study presents the mass of the lowermost stratosphere (LMS) and associated trends. It is correct that we define and compare different boundary surfaces enclosing the LMS.

In the following, we address all comments in detail (Reviewer's comments in italic, quotations of the corresponding revised text passages in blue).

Based on the suggestions of all reviewers, we have repeated our analyses of LMS structural changes with four modern reanalyses, namely ERA-Interim, MERRA-2, JRA-55 and JRA3Q, in addition to ERA5. The structure of the paper has essentially remained the same, although the individual sections have been expanded to include a comparison of the results between the reanalyses. We have also adapted the title accordingly: "Long-term changes in the thermodynamic structure of the lowermost stratosphere inferred from reanalysis data"

**Major comments**

*Overall, the manuscript reads well and sounds reasonable, and I find it a nice work. However, a concern arises from the beginning, whether the authors have used here ERA5 or ERA5.1. It is now well known that ERA5 has a cold bias in the lowermost stratosphere; therefore, its data are not reliable for some time in the region of the atmosphere that is the focus here. To address this issue, ERA5.1 was produced. The authors may have used the corrected ERA5.1 data; however, they should make it explicit. If they have used ERA5, there is a chance that the results here are not entirely valid (as they are based upon data known to be erroneous), and they should update the study using ERA5.1. On the other hand, if they have used ERA5.1, this should be made explicit in the text.*

Thank you for pointing out the cold bias in ERA5, rectified in ERA5.1. We agree that it is important to use ERA5.1 for our study of the lowermost stratosphere. All ERA5 analyses have been revised using ERA5.1 for the affected time period, i.e. 2000-2006. This revision has only led to minor alterations of the trend results.

*As some of the authors know (they have approached us in conferences to talk about it), some colleagues and I have proposed similar metrics to the ones here used for the UTLS region since years ago (joint PV and potential temperature changes), to determine the transition from tropics to the extratropical region (where the LMS mass is computed here), and regularly updated them. Although not published in a paper, they have been widely presented in SPARC workshops. Examples are:*

- *Añel, J. A., Gettelman, A., Castanheira, J. M., de la Torre, L. (2018) Tropical widening from isentropic and potential vorticity fields, SPARC OCTAV-UTLS Workshop. 7 - 9 November 2018, Mainz (Germany)*

- *Añel, J. A., Gettelman, A., Castanheira, J. M. (2015) Tropical widening from isentropic and PV fields. SPARC Regional Workshop on the Role of the Stratosphere in Climate Variability and Prediction, 12-13 January 2015, Granada (Spain)*

- *Añel, J. A., Gettelman, A., Castanheira, J. M. (2009) Tropical broadening vs. tropopause rising. "The Extratropical UTLS" SPARC Workshop, 19-22 October 2009, Boulder (CO, USA)*

*These metrics are based on the equivalent latitude of cross-points between isentropic lines and the PV field, so they are especially relevant for the discussion between lines 70 and 81 and the paragraphs after line 270 (which are the same as we have shown in the past) and Figure 9 (which is precisely the same kind of plot we have been presenting). Essentially, the results we have presented in the conferences have always matched those based on the tropopause break associated with the jet for all the reanalysis (ERA-Interim, NCEP2, JRA-55 and MERRA) and WACCM4 (the CCMI version with 66 vertical levels and in a configuration with 133 levels). It is comparable to the crossing between the 350 K isoline or 2 PVU and the thermal tropopause*

*used here. For the fairness and completeness of the discussion, it should be cited. The discussion in the Introduction in line 79, paragraphs 280-295, and the conclusions in line 367 are the right parts of the text to attribute the original idea and past results. Actually, I find it quite unfortunate that our works have not been cited in this submitted version. I have attached to this review one of our SPARC presentations to illustrate it.*

We did not mean to withhold any results and acknowledge your ideas and effort on the subject of tropical widening. Unfortunately, the mentioned talks are not easy to reference or to find for the interested reader. We therefore respectfully refrain from citing the mentioned work and instead refer to peer-reviewed literature, also referring to the ACP reference guidelines (`https://www.atmospheric-chemistry-and-physics.net/submission.html#references`).

*Additionally, I have read the manuscript several times, and I find a gap (maybe biased by my own scientific interests) in all the exposition and discussion related to the changes in the structure of the UTLS: the changes in the structure of the tropopause itself. I agree with the authors that mass changes are the relevant variable and that they have a criterion to delimit the region where it is computed. Additionally, they mention the overlapping of the tropical tropopause over the extratropical as an issue. However, from the point of view of the lapse-rate definition, it is clear that the region studied here is changing, which is evident in the broader area of vertical stability, fingerprinted by an increase in the numbers of multiple tropopauses (e.g. Castanheira et al. 2009), correlated with increasing UTLS baroclinicity. I think it is relevant to mention this around lines 96 and 225 and, if possible, to add in the manuscript some discussion on how the metrics presented here could be related to the changes in the vertical stability and the widening of the tropopause region.*

We agree that the increase of double tropopause frequency should be mentioned in our manuscript, as it can be related to changes in the thermodynamic structure of the UTLS. We have added according information, referring to the study by Castanheira et al. (2009).

L90-92: In addition to the general widening of the tropics, the frequency of double tropopause events, i.e. poleward excursions of the tropical above the extratropical tropopause, is found to have increased (Castanheira et al., 2009; Xian and Homeyer, 2019). This trend likely reflects an increase in baroclinicity in UTLS, driven by the GHG-induced climate change (Castanheira et al., 2009).

*Finally, using different periods (since 2000 and 1980) sometimes makes the text confusing. I do not see the point of beginning in 2000 and then extending the analysis back to the 1980s. Does it provide some fundamental new insight here? I doubt it. The authors could think about simply removing the part pre-2000.*

We take note of the experience that the discussion of different time periods can be confusing. Nevertheless, the main strength of the DLM (dynamic linear regression model) trend analysis is to infer potential trend reversal dates without prior specification. Even though our focus is on the time period after 1998, the non-linear DLM trends show interesting features also before this time. Especially the identification of trend reversal dates, as well as the specification of continuous trends, are helpful for interpreting the results after 1998. For example, considering both, the full time period 1979-2019 as well as changes after 1998, helps putting our results for the SH lapse rate tropopause pressure in context with trend studies focusing on different time periods.

**Minor comments**

*Line 42: citing Hoerling et al. (1991) about the potential vorticity and the tropopause is right. However, the numbers given by Hoerling et al. limit the location of the tropopause to 1-3 PVU, which has been proven too restrictive. Hoinka et al. (1999) have a good discussion, showing that 3.5 PVU approaches extratropical tropopause better. I recommend adding a citation to Hoinka et al. so that those readers without a profound knowledge of the topic have a more comprehensive and updated view of the issue of using the PV criterion to "find" the tropopause.*

In fact, Hoerling et al. (1991) compare tropopause definitions for PV threshold values between 1-5 PVU. Hoinka (1999) choose 3.5 PVU, referring to Hoerling et al. (1991).

*Line 43: when discussing the chemical tracers, I think it is fair to add the e90 by Prather et al., e.g. (2011) https://doi.org/10.1029/2010JD014939*

We acknowledge the existence of other tracers beside ozone, including e90, that allow for the definition of a chemical tropopause as shown by Prather et al. (2011). However, we aim to give a brief overview rather than a complete list of tropopause definitions in the mentioned text passage.

*Lines 88-94: I find this paragraph explicative and well-referenced. The authors mention that the issue of the BDC trends is an ongoing discussion. They refer to models, satellite data, and reanalysis. First, I would clarify in the text that Tegtmeier et al. (2020) refer to reanalysis. Then, I recommend citing a more up-to-date study, recently published by Sacha et al. (2024), which shows consistent results from models and uncertainties from reanalysis (https://doi.org/10.1029/2023GL105919).*

We appreciate the reference to the study by Šácha et al. (2024), which we have cited within the mentioned paragraph on BDC trends. Additionally, we have included the study by Zou et al. (2023), pointing out that the direction of tropical tropopause temperatures appears to have changed after 2005. Tegtmeier et al. (2020) report trends in the tropical tropopause layer for the period 1979-2005, comparing reanalysis data and observations.

L96-97: [...] According to Oberländer-Hayn et al. (2016) and Šácha et al. (2024), it is more precise to speak about a lifting of the circulation, which is connected to the tropopause expansion itself.
L105-110: [...] This is consistent with tropical lower stratospheric temperature trends close to zero within this period, inferred by Zou et al. (2023) from reanalyses data. The temperature reduction in the tropical tropopause region and at the cold point reported for the time period 1979–2005 by Tegtmeier et al. (2020) is consistent with increased tropical upwelling. However, different reanalyses often show a significant spread when compared, whether in terms of, e.g., temperature trends in the TTL region (e.g., Tegtmeier et al., 2020) or dynamical tropical upwelling (e.g., Šácha et al., 2024).

*I would remove the explanation on the Bayesian basis of the DLM, the paragraph beginning in line 98.*

We suppose this comment is related to line 168 of the original version of the manuscript. Its Bayesian nature distinguishes the DLM from other methods for time series analysis, e.g. multiple linear regression. We consider the short explanation on the DLM principles useful, especially for the understanding of the presented trend uncertainties.

*Also, the explanation of the accessibility to the DLM model code is already included in the "Code and data availability" section. Regarding this, a minor issue: GitHub is not a suitable repository to store assets from scientific research or papers; GitHub states it on its webpage and offers an integration with Zenodo to store code that needs long-term archival, as the one used in papers, providing a DOI for it. I strongly recommend copying the DLM code in Zenodo and citing it instead of the GitHub repository.*

Here, the goal was to show the source of the dlmmc which is available on github and can be referenced by Alsing (2019).

*Also, instead of making the LMS code available upon request (which outcome is never assured), I recommend depositing it in a permanent repository. ACP does not enforce this, but it is the usual practice in many other journals, including some of the EGU.*

We agree that publication of our code and data makes it easier for the scientific community to benefit from our code and data and to compare methods and results. We have therefore made the code for LMS mass calculation and trend estimation available on zenodo, together with the different LMS boundary fields (i.e. lapse rate tropopause, PPT10mean, PPTcp10mean as well as the cold point) for all five reanalyses. See `https://zenodo.org/records/13890232`.

*I think the colour scale for the DLM trend state in Fig. 3 should be improved. The discussion focuses on values lower than 7.5 hPa, and this is mostly yellow with independence of the values. It would be good if*

*the authors could provide a colour palette that helps to perceive the differences between 0 and 7.5 hPa.*

This is a valid point. We have improved the color scale.

*Lines 235-236: I would delete this mention of the polar vortex. Overall, the link between ozone recovery and its thermal effect and the material barrier that the polar vortex represents to latitudinal mixing is well-known and clear. However, I do not think it is actually relevant to the discussion here and only introduces some confusion.*

We think this is an interesting observation and always good to point out consistencies, also if relationships are well known.

*In the Conclusions, I would emphasise the "model" dependence of the results shown here for the LMS changes. There are some disagreements with previous works and probably with another reanalysis if it was checked.*

We are aware of the general strengths and limitations of reanalysis data and the differences between reanalysis datasets, specifically relevant for trend analyses. We realize that the manuscript was lacking a clear discussion on the uncertainties arising from this. Therefore, we have extended our analysis, comparing the previously presented results for ERA5 to three widely used reanalyses ERA-Interim, MERRA-2 and JRA-55 as well as the recently published reanalysis JRA3Q. In order to better address uncertainties of our findings, we point out robust features and discuss discrepancies across the different reanalyses.

**References**

Alsing, J.: dlmmc: Dynamical linear model regression for atmospheric time-series analysis, Journal of Open Source Software, 4, 1157, https://doi.org/10.21105/joss.01157, 2019.

Appenzeller, C., Holton, J. R., and Rosenlof, K. H.: Seasonal variation of mass transport across the tropopause, Journal of Geophysical Research: Atmospheres, 101, 15 071–15 078, https://doi.org/10.1029/96JD00821, 1996.

Castanheira, J. M., Añel, J. A., Marques, C. A. F., Antuña, J. C., Liberato, M. L. R., De La Torre, L., and Gimeno, L.: Increase of upper troposphere/lower stratosphere wave baroclinicity during the second half of the 20th century, Atmospheric Chemistry and Physics, 9, 9143–9153, https://doi.org/10.5194/acp-9-9143-2009, 2009.

Hoerling, M., Schaack, T., and Lenzen, A.: Global objective tropopause analysis, pp. 1816–1831, 1991.

Hoinka, K. P.: Temperature, Humidity, and Wind at the Global Tropopause, Monthly Weather Review, 127, 2248–2265, https://doi.org/10.1175/1520-0493(1999)127<2248:THAWAT>2.0.CO;2, 1999.

Prather, M. J., Zhu, X., Tang, Q., Hsu, J., and Neu, J. L.: An atmospheric chemist in search of the tropopause, Journal of Geophysical Research, 116, D04 306, https://doi.org/10.1029/2010JD014939, 2011.

Tegtmeier, S., Anstey, J., Davis, S., Dragani, R., Harada, Y., Ivanciu, I., Pilch Kedzierski, R., Krüger, K., Legras, B., Long, C., Wang, J. S., Wargan, K., and Wright, J. S.: Temperature and tropopause characteristics from reanalyses data in the tropical tropopause layer, Atmospheric Chemistry and Physics, 20, 753–770, https://doi.org/10.5194/acp-20-753-2020, 2020.

Zou, L., Hoffmann, L., Müller, R., and Spang, R.: Variability and trends of the tropical tropopause derived from a 1980–2021 multi-reanalysis assessment, Frontiers in Earth Science, 11, 1177 502, https://doi.org/10.3389/feart.2023.1177502, 2023.

Šácha, P., Zajíček, R., Kuchař, A., Eichinger, R., Pišoft, P., and Rieder, H. E.: Disentangling the Advective Brewer-Dobson Circulation Change, Geophysical Research Letters, 51, e2023GL105 919, https://doi.org/10.1029/2023GL105919, 2024.

---

## Author Comment (AC3)

**Reply to Reviewer 3**

We thank the Reviewer for the careful reading and constructive criticism of our manuscript. The comments and remarks have helped to improve the paper. We understand the main concerns raised by the Reviewer, regarding the discussion of uncertainties and justification of the choice of data and methodology. We have extended our analysis accordingly. In the following, we address all comments in further detail (reviewers comments in italic, quotations of the corresponding revised text passages in blue).

Based on the suggestions of all reviewers, we have repeated our analyses of LMS structural changes with four modern reanalyses, namely ERA-Interim, MERRA-2, JRA-55 and JRA3Q, in addition to ERA5. The structure of the paper has essentially remained the same, although the individual sections have been expanded to include a comparison of the results between the reanalyses. We have also adapted the title accordingly: "Long-term changes in the thermodynamic structure of the lowermost stratosphere inferred from reanalysis data".

**Major comments**

We realize that our manuscript can be improved by a more thorough discussion of uncertainties. In order to investigate the robustness of our findings, we have included a comparison of ERA5 with three widely used reanalyses, namely ERA-Interim, MERRA-2 and JRA-55 as well as the recently published JRA3Q data set. For the introduction and illustration of our metrics to assess the structure of the LMS, we use ERA5 before we generalize our findings to all other data sets. Furthermore, we have included a brief statement regarding the impact of the regressor variables on the trends. Beyond the aforementioned dependencies on data and regressors, we are confident that our method ensures a good treatment of uncertainties. One advantage of the DLM over other methods for time series analysis is the rigorous treatment of uncertainties in the data and the regression coefficients by simultaneous estimation of all model components. Further details are provided in the replies to the specific comments.

**Specific comments**

*The figures are clear and support the narrative. However, some figures (e.g., Figures 2 and 4) could benefit from putting individual lines into separate panels. Furthermore, the shading in these figures denotes the associated standard deviation, however, the statistical significance is tested on top of it. I would rather display 95% confidence or credible intervals (Šácha et al., 2024) straight away.*

Thank you for the suggestions. We have changed the figures accordingly, now presenting tropopause trends (Fig. 2), isentropic pressure trends (Fig. 5a–c) and pressure trends of the upper LMS boundaries (Fig. 9a–c) for ERA5 in individual panels. In addition to the mean and standard deviation, additional lighter shading now denotes the 5-95 percentile. In order to illustrate the reanalysis comparison without overloading the paper, we present the respective results in single panels. For the sake of clarity, we omit the presentation of uncertainty in these comparison figures but refer to the ERA5 plots as uncertainties are of the same order of magnitude across the reanalyses (Figs. 3, 5d–f and 9d–f).

*Explain the selection criteria for the reanalysis and its period since ERA5 goes beyond the year 1979*

We are aware of the potential and limitations of reanalysis data in general and of the individual reanalyses in particular, especially regarding long-term trend studies. Therefore, we agree that our study can benefit from a comparison of different reanalyses. While illustrating our metrics for LMS characterization with ERA5, we have included a comparison of the results in ERA-Interim, MERRA-2, JRA-55 and JRA3Q. We limit our trend analysis to the satellite era, starting in 1979. The time period 1979-2019 makes the different reanalyses reasonably comparable (MERRA-2 starting in 1980, 2018 being the last full year of ERA-Interim).

L134-135: For this study, ERA5 monthly mean data for the time period 1979–2019 has been used (Hersbach et al., 2020), 1979 marking the beginning of the satellite era. [...]
L144-146: In addition, we use monthly mean data from ERA-Interim (Dee et al., 2011), MERRA-2 (Gelaro et al., 2017), JRA-55 (Kobayashi et al., 2015) and JRA3Q (Kosaka et al., 2024) for the same time period as ERA5. However, ERA-Interim is only available until 2018 and the MERRA-2 time series begins in 1980.

*Why do you use only 2000 samples?*

This is a valid question since the number of Markov Chain Monte Carlo samples drawn from the posterior distribution of the DLM analysis is a choice left to the DLM user. In order to chose an appropriate DLM setup, we have performed sensitivity tests, comparing different numbers of DLM samples (e.g., 200, 1000, 2000, 5000, 10000) and their effect on the mean and spread of the DLM background trend state. The number of DLM samples does indeed influence the mean and spread of the resulting trends. A larger number of DLM samples does not necessarily reduce the spread, i.e. the uncertainty of the trend results but the trend with respect to, for example, latitude becomes smoother and thus more reliable. However, we consider the improvement from a number of 2000 to 10000 samples as negligible and not in proportion to the considerably higher computing effort. We therefore chose 2000 samples (plus 1000 warm-up samples). The same choice has also been made by Minganti et al. (2022). Karagodin-Doyennel et al. (2022) present DLM results for a sample number of only 200, while other studies compute 10000 samples (e.g., Laine et al., 2014; Šácha et al., 2024) and even 100000 samples (e.g., Ball et al., 2019).

L227-229: In this study, the DLM runs provide 2000 possible model state estimates after an additional 1000 samples that are considered as warm-up and discarded. Sensitivity experiments conducted as part of this study have shown that increasing the number of DLM samples (to e.g. 10 000) does not significantly improve the results, but comes at a considerable computational cost.

*Since the authors use other regressors, I would appreciate discussion of their impact and whether they contribute to reduce the uncertainty of the discussed trends.*

We conducted sensitivity test, running the DLM with and without regressors in order to check the robustness of the trends and their dependence on the regressors. However, our study does not aim to disentangle the contribution of different phenomena, represented by regression proxies, to the presented trends. Rather, we use the regressors for the sake of completeness and consistency with other studies on trends of UTLS characteristics (e.g. Seidel and Randel, 2006; Tegtmeier et al., 2020; Meng et al., 2021; Zou et al., 2023). However, we agree that a statement about the effect of the regressors was missing in the original manuscript. A short statement has now been added.

L215-220: [...] The same regressors have been used in different studies investigating changes of UTLS characteristics (e.g., Seidel and Randel, 2006; Meng et al., 2021; Tegtmeier et al., 2020; Zou et al., 2023). [...] In order to assess the robustness of the trends and the effect of the regressors, we conducted sensitivity tests in which the DLM was run with and without regressors. Overall, the results showed no strong dependency on the used regressors. However, cold point pressure trends, for example, become more significant when regressors are used (not shown).

*Using vector figures instead of raster ones may help to improve the quality of your publication.*

Thank you for the suggestion. We have changed all figures (except Figs. 3 and 5) to vector figures. (Figs. 3 and 5 become too large and are thus compressed.)

*I think the whole community would appreciate an adoption of Open Science approaches to allow the reproducing the extensive analysis in this study (e.g. Laken, 2016), especially when authors use DLMMC which has been made open. In particular, I would recommend any kind of willingness of the authors to publish the code allowing to reproduce the figures in the paper. There are multiple ways how to proceed, either to allow access upon request or via portals allowing to assignment Digital Object Identifier (DOI) to the research outputs, e.g. ZENODO. I think it could enhance the quality and reliability of this publication.*

We agree that publication of our code and data makes it easier for the scientific community to benefit from our code and data and to compare methods and results. We have therefore made the code for LMS mass calculation and trend estimation available on zenodo, together with the different LMS boundary fields (i.e. lapse rate tropopause, PPT10mean, PPTcp10mean as well as the cold point) for all five reanalyses.

See https://zenodo.org/records/13890232.

**Technical comments**

*L11 0.5° latitude per decade?*

It is 0.5°latitude between 1998-2019.

*L166 replace SOAD with SAOD and define*

Done.

**References**

Ball, W. T., Alsing, J., Staehelin, J., Davis, S. M., Froidevaux, L., and Peter, T.: Stratospheric ozone trends for 1985-2018: sensitivity to recent large variability, preprint, Gases/Remote Sensing/Stratosphere/Physics (physical properties and processes), https://doi.org/10.5194/acp-2019-243, 2019.

Karagodin-Doyennel, A., Rozanov, E., Sukhodolov, T., Egorova, T., Sedlacek, J., Ball, W., and Peter, T.: The historical ozone trends simulated with the SOCOLv4 and their comparison with observations and reanalyses, Atmospheric Chemistry and Physics, 22, 15 333–15 350, https://doi.org/10.5194/acp-22-15333-2022, 2022.

Laine, M., Latva-Pukkila, N., and Kyrölä, E.: Analysing time-varying trends in stratospheric ozone time series using the state space approach, Atmospheric Chemistry and Physics, 14, 9707–9725, https://doi.org/10.5194/acp-14-9707-2014, 2014.

Meng, L., Liu, J., Tarasick, D. W., Randel, W. J., Steiner, A. K., Wilhelmsen, H., Wang, L., and Haimberger, L.: Continuous rise of the tropopause in the Northern Hemisphere over 1980–2020, Science Advances, 7, eabi8065, https://doi.org/10.1126/sciadv.abi8065, 2021.

Minganti, D., Chabrillat, S., Errera, Q., Prignon, M., Kinnison, D. E., Garcia, R. R., Abalos, M., Alsing, J., Schneider, M., Smale, D., Jones, N., and Mahieu, E.: Evaluation of the $N_2O$ Rate of Change to Understand the Stratospheric Brewer-Dobson Circulation in a Chemistry-Climate Model, Journal of Geophysical Research: Atmospheres, 127, e2021JD036 390, https://doi.org/10.1029/2021JD036390, 2022.

Seidel, D. J. and Randel, W. J.: Variability and trends in the global tropopause estimated from radiosonde data, Journal of Geophysical Research, 111, D21 101, https://doi.org/10.1029/2006JD007363, 2006.

Tegtmeier, S., Anstey, J., Davis, S., Dragani, R., Harada, Y., Ivanciu, I., Pilch Kedzierski, R., Krüger, K., Legras, B., Long, C., Wang, J. S., Wargan, K., and Wright, J. S.: Temperature and tropopause characteristics from reanalyses data in the tropical tropopause layer, Atmospheric Chemistry and Physics, 20, 753–770, https://doi.org/10.5194/acp-20-753-2020, 2020.

Zou, L., Hoffmann, L., Müller, R., and Spang, R.: Variability and trends of the tropical tropopause derived from a 1980–2021 multi-reanalysis assessment, Frontiers in Earth Science, 11, 1177 502, https://doi.org/10.3389/feart.2023.1177502, 2023.

Šácha, P., Zajíček, R., Kuchař, A., Eichinger, R., Pišoft, P., and Rieder, H. E.: Disentangling the Advective Brewer-Dobson Circulation Change, Geophysical Research Letters, 51, e2023GL105 919, https://doi.org/10.1029/2023GL105919, 2024.

---

## Author Response (AR2)

**Reply to Editor**

Dear Petr Šácha,

Thank you for your feedback on our manuscript. We were pleased to hear that you consider only minor revisions to be necessary, which we hope we have now implemented to your approval. In the following, we address all comments in further detail (Editor's comments in italics, quotations of the corresponding revised text passages in blue. The line numbering of the editor's comments follow the previous tracked changes version, the replies relate to the current tracked changes version.)

**Editor's comments**

*Concerning the discussion phase, I see only one remaining complicated point - the comment from Dr. Añel about his long lasting, but unpublished contributions to the field. I have to make clear at this point that I am myself in a conflict of interest here, because I presented one of the presentations on behalf of the authors. Añel, J. A., Gettelman, A., Castanheira, J. M., de la Torre, L. (2018) Tropical widening from isentropic and potential vorticity fields, SPARC OCTAV-UTLS Workshop. 7 - 9 November 2018, Mainz (Germany). I am sorry for not realizing my potential conflict of interest before. Hence, concerning this point I refrain from making any editorial recommendations. My opinion is that this point could possibly be solved by acknowledging the discussions with Dr. Añel. I am sure that together with your co-authors you will find a balanced approach to this sensitive topic.*

As already stated in the reply to the comments of Dr. Añel, we acknowledge his contributions to the topic of tropical widening. However, we would still like to point out that the work in question is difficult to reference or find for the interested reader and refer again to the ACP reference guidelines (`https://www.atmospheric-chemistry-and-physics.net/submission.html#references`). We hope to have found a compromise with the following reference:

L386-387: Our metric and findings are similar to other metrics based on cross-points between isentropic and potential vorticity fields (J. Añel, personal communication).

**Editorial comments**

*1) At numerous places of the manuscript, the not shown statement is invoked (L238, L441) or the results are simply not shown (around L246 - sensitivity experiments with the DLM sample size, around L461 - LMS mass following Appenzeller et al). At all instances, those results are important for the study and should be shown at least in the Appendix.*

We had refrained from presenting the corresponding results in order to avoid an extensive appendix. However, we acknowledge that the results can be important for the study. Regarding L238 (now L224, trend results with and without regressors): We have extended Fig. 2 so that it now shows the ERA5 tropopause trends with (color) and without regressors (grey). Moreover, the LMS mass trends across the reanalyses for the DLM run without regressors can be found in the appendix (Fig. A2). These examples show that the use of regressors has no strong effect on the trend results, as stated in the text:

L223-224: Overall, the results showed no strong dependency on the used regressors. However, tropical tropopause pressure trends, for example, become more significant when regressors are used (Figs. 2, A2).

Regarding 441 (now L416-418, location of boundary surfaces across reanalyses): We have added a figure illustrating the differences in the location between the individual LMS boundary surfaces in the reanalyses (A6). Also, we have refined the corresponding statement in the text:

L416-420: The MERRA-2 LMS mass deviates the most from the other reanalyses, generally showing smaller values but the relative differences between the data sets are smaller than $10\%$ in the NH and smaller than $20\%$ in the SH. The variations of LMS mass across the reanalyses can be primarily attributed to the differences in tropopause location, determining the lower LMS boundary as well as the dynamical

upper boundaries (Fig. A6).

Regarding L246 (now L232-234, DLM sample number): We have added an example comparing DLM trend results for different sample numbers in the appendix (Fig. A1).

L232-234: Sensitivity experiments conducted as part of this study have shown that increasing the number of DLM samples (to e.g. 10 000) does not significantly improve the results, but comes at a considerable computational cost (see for example Fig. A1).

Regarding L461 (now L437-438, LMS mass similar to Appenzeller et al. (1996)): We considered adding the LMS mass values calculated for LMS boundaries similar to Appenzeller et al. (1996), to Tab. 2. However, as mentioned in L437-438, we shifted the ERA5 $2\,\mathrm{PVU}$ iso-surface by $60\,\mathrm{hPa}$ to approximately match the UKMO $2\,\mathrm{PVU}$ surface illustrated in Appenzeller et al. (1996) Fig. 3. This is intended as proof of concept but does not represent a physical LMS boundary surface. Neither did we use the same data asAppenzeller et al. (1996). Therefore, we decided not to include the respective LMS mass in Tab. 2.

*2) Structure of the paper:*

*2)A) Fig. 13 is absolutely misplaced.*

We recognize that the placement of Fig. 13 was not ideal. We have moved the figure (Fig. 13, now Fig. 4) up to Sect. 3.1. where it is first mentioned.

*B) Many subplots are not mentioned in the text (Fig. 3b, d; Fig. 4b, c) and some figures are only introduced in the paper and not discussed afterwards (Fig. 7). I recommend you to either expand the description of the results or move the related figures (or subplots) to the supplement (Appendix)?*

Regarding Fig. 3: We adjusted the referencing in the text and the subplots of Fig. 3 are now specifically addressed in the text:

L257-258: All considered reanalyses agree on negative pressure trends at the tropical lapse rate tropopause and the cold point (Fig. 3a and b), whereas potential temperature trends differ in sign (Fig. 3c and d).

Regarding Fig. 4 (now Fig. 5): The subplots Fig. 4b and c (now Fig 4 and A3) were shown for the sake of completeness. We realize that the additional subplots, which are not discussed in the text, can be moved to the appendix.

L276-277: The DLM trend state of $2\,\mathrm{PVU}$ and cold point pressure can be found in the appendix (Fig. A3).

Regarding Fig. 7 (now Fig. 8): This figure is intended to provide an overview of the location of the boundary surfaces. Unlike Fig. 1, Fig 7 (now Fig. 8) it is not a schematic, but illustrates the zonal mean, temporal mean pressure of the LMS boundaries as used in this study for the concrete example of ERA5. Another purpose of the figure is to justify the definition of the lateral LMS boundary. We realize that some references to the figure, especially corresponding to the latter topic, were lacking. They have now been added.

L335-336: The location of the different boundary surfaces used for LMS definition are displayed in Fig. 8 using ERA5 as an example.

L366-368: Specifically, the lapse rate tropopause and the $2\,\mathrm{PVU}$ surface at $350\,\mathrm{K}$ are close to the isentropic PV-gradient tropopause, which marks a clear transport barrier between the tropical troposphere and the extratropical lower stratosphere, i.e. the LMS (Kunz et al., 2011; Turhal et al., 2024).

L407-409: Since the $2\,\mathrm{PVU}$ tropopause is located at significantly higher pressure than the lapse rate tropopause (fig. 8), the LMS mass between $2\,\mathrm{PVU}$ and the respective upper boundary is 1.6-1.8 times

larger than the mass with respect to the thermal tropopause.

*C) L374-L394 In this part of the paper you keep jumping between Figs. 8 and 9 and also Fig. 13 is suddenly invoked. Can you rewrite the text or reorganize the figures for a better readability?*

We realized that the presentation of Figs. 8 and 9 (now 9 and 10) did not match the corresponding text. In the text, the relationship between trends of the upper LMS boundaries and the potential temperature at the tropical tropopause are first discussed, using the example of ERA5. Afterwards, the reanalyses are compared. The figures have been reorganized in a consistent manner (Figs. 9 and 10). The text has been adapted accordingly and refined in some places. Note the new references in L338-345 and:

L349-357: Like in ERA5, a significant rise of the PPT10mean surface in the extratropics is also evident in ERA-I and MERRA2 (Fig. 10b). This can be attributed the increase of tropical tropopause potential temperatures, which define the isentropes corresponding to PPT10mean (Fig. 10d). JRA-55 and JRA3Q on, the other hand, suggest the opposite (Fig. 10b and d). At the cold point, ERA5 and ERA-Interim agree on continuously increasing potential temperatures between 1998–2019, whereas the JRA data sets show mostly decreasing potential temperatures and MERRA-2 almost no trend at all (Fig. 10e). This is reflected in the trends of the PPTcp10mean surface (Fig. 10c). The contrasting behavior of the reanalyses can at least partly be attributed to the respective temperature trends in the TTL region.

*3) Lifting of the isobars, isentropes and the tropopause.*
*I think that you should approach the issue of different vertical shifts either more rigorously (e.g. analytically using the transformation of coordinates as eq. 1 in Šácha et al., 2019, ACP - note the corrigendum here) or maybe a visualization of different vertical shift rates of the tropopause and surrounding isobars and isentropes (e.g. similar to Fig. 4 in Šácha et al., 2019, ACP) would help the reader to orient in this field.*

The LMS mass trends are supposed to be the core of this study. As the LMS mass is defined by the pressure difference between the upper and lower boundaries, we set focus on pressure trends, i.e. pressure as the vertical axis. Changes of the location of the tropopause and isentropes are presented to improve the understanding of the observed LMS mass trends and to illustrate the role of the respective LMS boundaries. For context, we decided to include geopotential height trends of the tropopause (Fig. 2) and isentropes in ERA5 (A4). However, we do not aim to rigorously study the relationship between geometric height and pressure at this point. We realized that at some places in the manuscript, arguments referring to pressure and height reference systems had still been mixed. We have revised the wording in the relevant places:

L407-412: Since the $2\,\mathrm{PVU}$ tropopause is located at significantly higher pressure than the lapse rate tropopause (Fig. 8), the LMS mass between $2\,\mathrm{PVU}$ and the respective upper boundary is 1.6–1.8 times larger than the mass with respect to the thermal tropopause. The LMS mass with respect to the different upper boundary surfaces varies by less than $15\,\%$ for the same lower boundary and hemisphere.

L536-539: In general, the vertical boundaries dominate LMS mass changes. However, tropical width changes can have a considerable impact on LMS mass due to the area decrease from low to high latitudes.

**Technical comments**

*L8-9: The sentence "This is consistent with a strengthening of the Brewer-Dobson circulation." should be deleted from the abstract, because the connection is not clear at this point.*

We have deleted the sentence from the abstract.

*L40-41 If you are aware of any paper documenting the seasonal cycle in stratospheric mass, you should reference it here. Otherwise do not make statements about stratospheric mass.*

Appenzeller et al. (1996), as cited in L39 (previously L41), present the mass of the lowermost stratosphere as well as total stratospheric mass, including the seasonal cycle of both. Estimated from Figs. 5a and c in Appenzeller et al. (1996), the proportion of LMS mass to total stratospheric mass is roughly 30–$50\,\%$. From our own calculations, we estimated the contribution of LMS mass at ca. $20\,\%$ of the total

stratospheric mass. This number was derived from approximating the total stratospheric mass between the tropopause and $1\,\mathrm{hPa}$. We adjusted the wording as follows:

L37-38: From Fig. 5 in Appenzeller et al. (1996), the LMS mass can be estimated to about 30–50 % of the total stratospheric mass.

L413-414: If the upper edge of the stratosphere is approximated at $1\,\mathrm{hPa}$, the proportion of LMS mass to total stratospheric mass is about 20 %.

L541-543: LMS mass changes are important to consider because the LMS mass makes up a considerable amount of the stratospheric column, in our case approximately 20 %, and therefore contains a substantial fraction of, for example, column ozone.

L74.. despite the warming of the stratosphere..please rephrase to make clear that contrasting effects can be anticipated from the ozone recovery and GHG cooling in the stratosphere.

We hope the amended wording makes the contrasting effects clearer:

L69-73: [...] increasing greenhouse gas emissions and the resulting warming of the troposphere and cooling of the stratosphere. The tropospheric warming was found to cause a persistent lifting of the tropopause, even as contrasting temperature effects are expected in the stratosphere, where ozone recovery is causing warming while increasing GHG load is exerting a cooling influence (Pisoft et al., 2021; Meng et al., 2021).

L144-145 ..For this study, ERA5 monthly mean data for the time period 1979–2019 has been used (Hersbach et al., 2020), 1979 marking the beginning of the satellite era...This sentence should be deleted because this information is already covered by the preceding paragraph in the revised version.

The first paragraph in section 2.1 very briefly introduces the data used, listing the different reanalysis data sets. The choice of time period and monthly mean data is only mentioned in the second paragraph. Therefore, we think the mentioned sentence is important. We have changed the sentence slightly in the hope of achieving a better reading flow:

L137-138. We use ERA5 monthly mean data for the time period 1979–2019 (Hersbach et al., 2020), 1979 marking the beginning of the satellite era.

Eq.1 - This formula originally from Appenzeller et al. (1996) does not take into account the lateral boundaries? I suspect that for your case the bounds of integration should be modified.

Thank you for pointing this out. The variable lateral boundaries are indeed important to be included in Eq. 1:

L205-208:

$$M(t) = \int_0^{2\pi} \int_{\Phi_1(\lambda,t)}^{\Phi_2(\lambda,t)} \int_{p_1(\lambda,\Phi,t)}^{p_2(\lambda,\Phi,t)} -\frac{1}{g} dp \cos\Phi d\Phi d\lambda, \tag{1}$$

$g$ is the gravity constant, $\lambda$ is the longitude, $\Phi$ is the latitude and $dp$ is the pressure difference. $\Phi_1(\lambda,t)$ and $\Phi_2(\lambda,t)$ denote the lateral boundaries, defined by the intersection between the tropopause and the $350\,\mathrm{K}$ isentrope for every longitude and time step.

L275-276 - In Fig. 3, it seems to me that for cold point the reanalyses differ in both pressure and potential temperature trend?

The magnitude of the cold point trends differs across the reanalyses but the sign is the same. The latter is what we want to emphasize here. We have changed the wording, hoping to make the statement clearer:

L257-258: All considered reanalyses agree on negative pressure trends at the tropical lapse rate tropopause and the cold point (Fig. 3a and b), whereas potential temperature trends differ in sign (Fig. 3c and d).

*Figs. 4 and 6 and the corresponding discussion in the text: You write about negative trends but the color scale starts at zero?*

Figs. 4 and 6, now Figs. 5 and 7, illustrate the non-linearity of DLM trends. As stated in the figure captions, Fig. 5 (7) shows the DLM trend state time series, i.e. the hidden mean state of tropopause (isentropic) pressure relative to the minimum at every latitude. The point in time at which the trend state at one latitude reaches its minimum is indicated by a black dot. The pressure trend state is higher before reaching the minimum, i.e. decreasing. Accordingly, the pressure trend state is increasing after having reached the minimum. We have amended the wording to make the interpretation of Figs. 5 and 7 clearer:

L272-276: The non-linear DLM trend analysis for the entire time series 1979–2019 reveals a trend reversal of the ERA5 lapse rate tropopause pressure around the year 2000. This trend reversal is evidenced by the fact that the DLM pressure trend state reaches its minimum around the year 2000 (Fig. 5). Accordingly, the DLM results suggest decreasing pressure of the ERA5 SH mid-latitude lapse rate tropopause before the year 2000 and increasing pressure thereafter.

L311-312: This trend reversal is evident from the minimum in the DLM trend state (Fig. 7).

*Fig. 5 and the corresponding discussion around L315 - Except maybe in the tropics the trends are very uncertain, but the discussion does not reflect this. You state this only at L319. But, the notion of significance should be the first information the reader gets.*

We agree that the uncertainty of the trends should be mentioned along with their magnitude. We have added the missing information:

L290-292: The average pressure trends range between $-0.3$ to $+0.6\,\mathrm{hPa/decade}$, which corresponds to an absolute pressure change of up to $1\,\mathrm{hPa}$ during the 21 year period, albeit exhibiting large uncertainties, especially in the extratropics.

*L318 - "...accompanied by a rise of the respective potential temperature.." - you should specify what is the vertical coordinate relative to which the rise is diagnosed (height, pressure...)*

We have specified the vertical coordinates in the mentioned paragraph:

L293-296: [...] which is accompanied by a rise of the respective potential temperature iso-surfaces in geopotential height (Fig. A4a). The isentropic pressure trends [...]. However, while the $380\,\mathrm{K}$ isentrope in ERA5 and MERRA-2 shows no pressure trend in the tropics and subtropics [...].

*L320-L323 - I like very much your discussion on the possible connection of the pressure of isentrops with BDC. Don't you want to devote more space in the manuscript to this?*
*and*
*Paragraph starting with L324 on isentropic pressure trend - Here again, BDC strength (tropical upwelling) can be a part of the story?*

The discussion regarding the relationship between isentropic pressure trends and BDC changes is taken up again and deepened in lines L320-325 (previously L344-349). We recognize that a reference to the further discussion of the topic in the following paragraph (considering the non-linear DLM trends) can be helpful for the reader. We have added such a reference, linking the subsequent paragraphs:

L305-306: While the average pressure changes discussed in this paragraph provide valuable insights, examining the full, non-linear DLM trend state time series can offer a more comprehensive view of the physical mechanisms driving trend evolution. (L309-310 are therefore no longer necessary and have been deleted.)

*Discussion around L340 - Negative trends can not be seen in Fig. 6. Moreover, why do you expect this kind of mismatch between the rise of isobars and isentropes for diabatic heating? Can you give some*

*reasoning (ideally analytically)?*

As explained above, Figs. 5 and 7 (previously 4 and 6) illustrate the non-linear DLM trend states. The color shading shows the pressure difference with respect to the respective minimum for every latitude bin and every time step. The minima are indicated by the black dots. At every latitude, pressure is decreasing until reaching the minimum and increasing thereafter. This way, Fig. 7, reveals continuously negative isentropic pressure trends in the tropics and extratropics above $380\,\mathrm{K}$.

The relationship between isentropic pressure and temperature simply follows from the definition of potential temperature: $\Theta = T * (\frac{p_0}{p})^\kappa$. An isentrope, i.e. $\Theta = const.$, is shifted to higher pressure if the temperature increases.

L315-316: This relationship between lower stratospheric temperatures and isentropic pressure directly follows from the definition of potential temperature.

*L346-347 "Diabatic cooling of the tropical lower stratosphere results from increasing GHG load in the troposphere and a continuous decline of lower stratospheric ozone." - Stratospheric cooling is also a function of stratospheric GHGs. See the paper of Vallis et al. (2015, QJRMS) for the physical reasoning. But, it is hard to extrapolate this general knowledge to the tropical LS trends in particular.*

Thank you for the reference. We agree that beside tropospheric GHG load and stratospheric ozone concentrations, (lower) stratospheric temperature changes also depend on stratospheric GHG concentrations. Accordingly, we have changed the wording from "GHG load in the troposphere" to „GHG load in the atmosphere" and added references to the mentioned study by Vallis et al. (2015) as well as an earlier study by Ramaswamy et al. (2001). Furthermore, we agree that these are general relationships and not specific to the tropics. We have therefore relativised the statement, using the wording "can be expected" instead of "result".

L322-325: [...] diabatic cooling of the tropical lower stratosphere can be expected from increasing GHG load in the atmosphere and a continuous decline of lower stratospheric ozone (e.g., Ramaswamy et al., 2001; Vallis et al., 2015). Such a cooling of the lower stratosphere [...].

*P18 - I do not think that the footnote is neccessary.*

We have removed the footnote.

*L395 I miss here some provisional wrap-up of the results that will inform the following analysis. Like based on this and this, we choose the following two upper boundary definitions..etc. Sentences like this would enhance the readability of the text.*

We agree that the readability benefits from a brief conclusion of the preceding paragraph (dealing with the upper LMS boundaries), which we have now added. Furthermore, we have added a short outlook on the mass trend, to make the context clear.

L360-362: For the LMS mass analysis in Sect. 3.3.2 and 3.3.3, we use $380\,\mathrm{K}$, PPT10mean and PTcp10mean as upper LMS boundaries and compare their respective effects on the mass. The different trends of the iso-surfaces discussed here are reflected in the LMS mass trends.

*Starting at L418 - Here you forget to mention the behavior of ERAI, MERRA2 and partly JRA55, which is very different, and possible causes of this (different handling of O3 ?)*

We agree that the mentioned discussion was a bit vague. We now refer more clearly to the differences across the reanalyses. However, we refrain from speculating on the reasons for the differences across the reanalyses, as we do in the rest of the manuscript. The objective of our study is to examine the physical changes in the LMS. The reanalysis comparison is a useful tool for assessing the robustness of our findings and highlighting differences across the reanalyses. It is, however, not within the scope of our study to elucidate the (technical) reasons for discrepancies between the reanalyses. In the conclusions, we refer to

Fujiwara et al. (2022) and Fujiwara et al. (2024) (L529-532).

L388-395: In the SH, on the other hand, the lateral tendencies of the intersections between tropopause and isentropes are less clear. For the lapse rate tropopause at $350\,\mathrm{K}$, all reanalyses except JRA3Q agree on an equatorward trend, however differing in magnitude and statistical significance. In contrast to the lapse rate tropopause, the $2\,\mathrm{PVU}$ dynamical tropopause in ERA5 shows a poleward trend at $350\,\mathrm{K}$ (Fig. A5a). The general shape of the lateral tropopause trends with respect to potential temperature, however, is similar in most reanalyses [...].

*L430-431 - Maybe I missed it in the preceding text, but why do you choose in the end these two combinations of the LMS boundaries?*

In fact, we use the combinations lrtp-380K, lrtp-PPT10mean and lrtp-PPTcp10mean for all reanalyses. For ERA5, we use all combinations 2PVU-* in addition. Here, the combinations lrtp-PPT10mean and 2PVU-PPTcp10mean are mentioned, because these result in the minimum and maximum average LMS mass. We tried to make this more clear in the text.

L404-407: The mean LMS mass in ERA5 between all considered boundary surfaces on both hemispheres is listed in Tab. 2. In ERA5, the average global LMS mass between the different boundary surfaces (lrtp, $2\,\mathrm{PVU}$ and $380\,\mathrm{K}$, PPT10mean, PPTcp10mean) ranges from $0.99 \pm 0.28 \cdot 10^{17}$ kg (resulting from lrtp-PPT10mean) to $1.89 \pm 0.37 \cdot 10^{17}$ kg (resulting from $2\,\mathrm{PVU}$-PPTcp10mean) within the time period 1979–2019.

*L436 "The mean LMS mass in ERA5 between all considered boundary surfaces on both hemispheres is listed in Tab. 2." - I recommend this to be the starting sentence of the section 3.3.2*

Thank you for the suggestion. Section 3.3.2 now begins with the mentioned sentence.

*L461 - I suggest adding the LMS mass following Appenzeller to Table 2.*

See comment above. We refrained from adding the mentioned mass to Tab. 2 because it was calculated as proof of concept and does not represent an actual result.

*L469 For the ERA 5-> Only for ERA5 ...*

We have changed the wording to:

L444: For the LMS in ERA5 bounded with $380\,\mathrm{K}$ [...].

*Around L475 - The contributions of the mechanism 1)-3) behind the LMS mass changes can be quantified by differentiating the eq. 1 in a similar way as Šácha et al. (2024, GRL) differentiates the definition of the net tropical upwelling.*

We agree that the discussion of the LMS mass trends benefits from a quantification of the contributions from the different boundary surfaces. In order to asses these contributions, we conducted sensitivity studies, allowing only one boundary surface to evolve in time while the other two had been fixed. More specific, for one boundary surface, we used the same 4D pressure filed as before, while we repeated the first year of the time series (1979) for the other two. We present the results of this sensitivity study in Fig. 13 to confirm the statements in L449-453. We realized, that these statements needed some refinement and changed their order.

L449-458: This suggests three things: 1) The decreasing NH LMS mass for a fixed $380\,\mathrm{K}$ boundary is largely due to the rising NH extratropical tropopause. 2) The "dynamical" upper boundary surfaces based on the tropical tropopause are indeed able to largely compensate for the tropopause rise in ERA5. 3) Locally, a considerable proportion of this NH LMS mass decline can be attributed to the poleward trend of the lateral boundary, which is the result of an expansion of the tropics, while the hemispheric effect is rather small.

Fig. 13 confirms these observations and helps to quantify the contributions of the respective boundary surfaces to the hemispheric LMS mass changes. The figure compares LMS mass changes for an LMS where all boundary surfaces are evolving in time (as before, Fig. 12) with an LMS with only one boundary surface evolving in time ("floating"), while the other two are fixed (here, the first year of the time series is repeated).

L473-475: [...] highlights the impact of the upper boundary surface, which is confirmed again in Fig. 13b.

*L494 - "In the SH, LMS mass trends differ more strongly between the reanalyses than in the NH (Fig. 12)..." - I do not see by eye that the differences between reanalyses are greater in SH than in NH.*

We recognize that Fig. 12 (now Fig. 14) can be improved in order to highlight the mentioned differences. Furthermore, we specified that we are especially referring to the mean trends and the LMS mass between lrtp-380K and lrtp-PPT10mean.

L476-477: In the SH, the mean LMS mass trends differ more strongly between the reanalyses than in the NH (Fig. 14g–l), especially for the LMS bounded by lrtp-$380\,\mathrm{K}$ and lrtp-PPT10mean.

*L536 "...suggests a steepening of the tropopause break." - If this feature is that important to be highlighted in the conclusions, what prevents you to analyze it in the paper instead of only speculating about its role?*

We realize that our wording may not have been sufficiently precise here. The lateral tropopause trends with respect to potential temperature, as presented and discussed in Sect. 3.3.1 suggest a steepening of the subtropical tropopause, not the tropopause break in particular. We have changed the wording accordingly. In general, we believe that this result is worth mentioning in the conclusions.

L395-396: The different magnitudes of the lateral trends for tropopause intersections with different isentropes suggests a shape change of the subtropical tropopause.

L398-400: Furthermore, such a steepening of the subtropical tropopause, including the tropopause break, could be an indication of a strengthened BDC (Birner, 2010b).

L513-518: The different magnitude of the lateral trends with respect to the potential temperature on both hemispheres suggests a steepening of the subtropical tropopause. [...] Such a steepening of the subtropical tropopause, including the tropopause break, can be associated with an intensification of the SH subtropical jet (Manney and Hegglin, 2018; Maher et al., 2020) and could be linked to a strengthening of the BDC (Birner, 2010b).

*L550-L551 - Hints can be found in Fujiwara et al. (2022). - Also, in Fujiwara et al. (2024, ACP) it can be seen that JRA55 differs from other reanalyses in terms of SW and LW heating in the upper troposphere in DJF and JJA.*

Thank you for the reference to the study of Fujiwara et al. (2024), which we have now included. However, we want to mention again that it is beyond the scope of our study to assess the reasons for the differences across the reanalyses in more depth.

L529-531: Hints can be found in Fujiwara et al. (2022) and Fujiwara et al. (2024), suggesting relationships between differences in tropical lower stratospheric temperature and radiative heating, related to ozone concentrations, among other factors.

*L840 The references Škerlak and Šácha should be placed after S and before T.*

We have fixed this error.

**References**

Appenzeller, C., Holton, J. R., and Rosenlof, K. H.: Seasonal variation of mass transport across the tropopause, Journal of Geophysical Research: Atmospheres, 101, 15 071–15 078, https://doi.org/10.1029/96JD00821, 1996.

Fujiwara, M., Manney, G. L., Gray, L. J., and Wright, J. S.: SPARC Reanalysis Intercomparison Project (S-RIP) Final Report, Tech. rep., `https://elib.dlr.de/148623/`, 10th assessment report of the SPARC project, published by the International Project Office at DLR-IPA. also: WCRP Report 6/2021, 2022.

Fujiwara, M., Martineau, P., Wright, J. S., Abalos, M., Šácha, P., Kawatani, Y., Davis, S. M., Birner, T., and Monge-Sanz, B. M.: Climatology of the terms and variables of transformed Eulerian-mean (TEM) equations from multiple reanalyses: MERRA-2, JRA-55, ERA-Interim, and CFSR, Atmospheric Chemistry and Physics, 24, 7873–7898, https://doi.org/10.5194/acp-24-7873-2024, 2024.

Ramaswamy, V., Chanin, M., Angell, J., Barnett, J., Gaffen, D., Gelman, M., Keckhut, P., Koshelkov, Y., Labitzke, K., Lin, J. R., O'Neill, A., Nash, J., Randel, W., Rood, R., Shine, K., Shiotani, M., and Swinbank, R.: Stratospheric temperature trends: Observations and model simulations, Reviews of Geophysics, 39, 71–122, https://doi.org/10.1029/1999RG000065, 2001.

Vallis, G. K., Zurita-Gotor, P., Cairns, C., and Kidston, J.: Response of the large-scale structure of the atmosphere to global warming, Quarterly Journal of the Royal Meteorological Society, 141, 1479–1501, https://doi.org/10.1002/qj.2456, 2015.